# Process Algebraic Approach for Probabilistic Verification of Safety and Security Requirements of Smart IoT (Internet of Things) Systems in Digital Twin

**Junsup Song** [1], **Sunghyun Lee** [1], **Dimitris Karagiannis** [2] **and Moonkun Lee** [1,*]

1   Department of Computer Science and Engineering, Jeonbuk National University, Jeonju 561-756, Republic of Korea; junsup@jbnu.ac.kr (J.S.); shlee1@jbnu.ac.kr (S.L.)
2   Research Group Knowledge Engineering, University of Vienna, 1010 Vienna, Austria; dimitris.karagiannis@dke.univie.ac.at
*   Correspondence: moonkun@jbnu.ac.kr

**Abstract:** Process algebra can be considered one of the most practical formal methods for modeling Smart IoT Systems in Digital Twin, since each IoT device in the systems can be considered as a process. Further, some of the algebras are applied to predict the behavior of the systems. For example, PALOMA (Process Algebra for Located Markovian Agents) and PACSR (Probabilistic Algebra of Communicating Shared Resources) process algebras are designed to predict the behavior of IoT Systems with probability on choice operations. However, there is a lack of analytical methods in the algebras to predict the nondeterministic behavior of the systems. Further, there is no control mechanism to handle undesirable nondeterministic behavior of the systems. In order to overcome these limitations, this paper proposes a new process algebra, called dTP-Calculus, which can be used (1) to specify the nondeterministic behavior of the systems with static probability, (2) verify the safety and security requirements of the nondeterministic behavior with probability requirements, and (3) control undesirable nondeterministic behavior with dynamic probability. To demonstrate the feasibility and practicality of the approach, the SAVE (Specification, Analysis, Verification, Evaluation) tool has been developed on the ADOxx Meta-Modeling Platform and applied to a SEMS (Smart Emergency Medical Service) example. In addition, a miniature digital twin system for the SEMS example was constructed and applied to the SAVE tool as a proof of concept for Digital Twin. It shows that the approach with dTP-Calculus on the tool can be very efficient and effective for Smart IoT Systems in Digital Twin.

**Keywords:** smart IoT; digital twin; formal method; process algebra; dTP-Calculus; probability; SAVE; ADOxx



## 1. Introduction

### 1.1. Digital Twin

*Digital Twin* [1], known as Digital Mirroring, is a new technology that models objects in real world with ICT and enables them and their operations interact with their digital models [2,3]. In that perspective, Digital Twin not only receives information from the objects in the real world but also controls them, beyond simple modeling them as digital entities [4–6].

Due to that perspective, recently, Digital Twin has become one of the main innovative topics for Big Data, AI, IoT, Smart Systems, etc., in the areas of manufacturing, medical service, aerospace, defense, agriculture, and so on [3,7–12]. Further, Digital Twin is considered a subset of a CPS (Cyber-Physical System) with high fidelity [13]. More specifically, it is used to specify, verify, and control various requirements of complex physical systems, such as Smart Systems, in various industries.

Consequently, in recent years, Smart IoT has been merging into Digital Twin to enhance the smart functionalities of Smart Computing based Big Data and AI on IoT [14]. For example, Smart IoT Medical Systems have utilized smart IoT devices in order to transfer the health-related information of patients in real-time to medical centers through the digital twin systems [15]. If the devices work improperly, the patients' lives may be in danger. Therefore, it is necessary to verify the safe and secure functionalities or requirements of the devices and systems in Digital Twin prior to the physical implementation of the devices in the systems. More specifically, Digital Twin should provide a mechanism to predict the unexpected behavior of the systems, not satisfying such functionalities and requirements, and control the behavior so that all the functionalities and requirements are satisfied.

### 1.2. Process Algebra

The most common approaches for the verification of the safety and security requirements of Digital Twin are based on formal methods, such as, logic, state machine, and process algebra [16,17]. Among these, as stated in the abstract, the approach based on process algebra is most promising: each IoT device can be considered as a process Twin with the properties of real-time, periodicity, control, interactivity, mobility, distribution, etc. [18,19].

Further, some of these algebras are applied to predict the behavior of the systems. For example, PALOMA (Process Algebra for Located Markovian Agents) and PACSR (Probabilistic Algebra of Communicating Shared Resources) process algebras are designed to predict the behavior of IoT Systems with probability on choice operations [18,19]. However, there is a lack of analytical methods in the algebras to predict the nondeterministic behavior of the systems, since the prediction of their behavior can be expressed as unconditional nondeterministic choice operations of the process algebras, based on the static probabilistic model. More specifically, PACSR exclusively relies on discrete probability models, and PALOMA solely utilizes exponential probability distribution models. They are not suitable for the analysis of dynamic probability models to control nondeterministic choice operations. Further, there is no control mechanism to fix undesirable nondeterministic behavior of the systems for both algebras.

### 1.3. Approach

To overcome these limitations, this paper presents a new formal method, called dTP-Calculus, which is extended from dT-Calculus [18,19]. It provides a method to specify, analyze, and control the dynamic behavior of Smart IoT Systems in Digital Twin, based on static and dynamic probability. Further, dTP-Calculus can be used to specify and verify probabilistic safety and security requirements of Smart IoT Systems in Digital Twin, based on probability.

Figure 1 shows the main steps and points of the approach introduced in the paper, based on dTP-Calculus, as follows. Note that it is one of the most typical approaches in the formal-methods-based software engineering methodology [20]:

(1)    The probabilistic operational requirements of a Smart IoT System are specified with dTP-Calculus by defining a system model and the process models for all the processes in the system model. Specifically, the nondeterministic behavior of the system is specified with the probabilistic choice operation of each process with static probability.

(2)    A probabilistic execution model for the system is determined, and all the execution paths of the system are generated. Particularly, the nondeterministic behavior of the system is generated by the composition of the probabilistic choice operations of the processes.

(3)    Each path of the probabilistic execution model is simulated, based on the semantics of the calculus, and its output is generated in the form of the GTS (Geo-Temporal Space) model. Specifically, the nondeterministic behavior of the system is quantitatively measured in the GTS model and expressed with the quantitative values of the probabilities composed by the probabilistic choice operations of the processes.

(4) The safety and security requirements of the system are specified with GTS Visual Logic on the GTS generated from the previous step. Particularly, the probability of the satisfiability of the requirements is measured with the composed probability from the above (3).

(5) The safety and security requirements are analyzed and verified on the GTS. Specifically, the probability of the satisfiability of the requirements is verified with the probability requirements.

(6) If the probability requirements are not satisfied, the probabilities for the requirements are re-generated, and new criteria to satisfy the requirements are reset with dynamic probability.

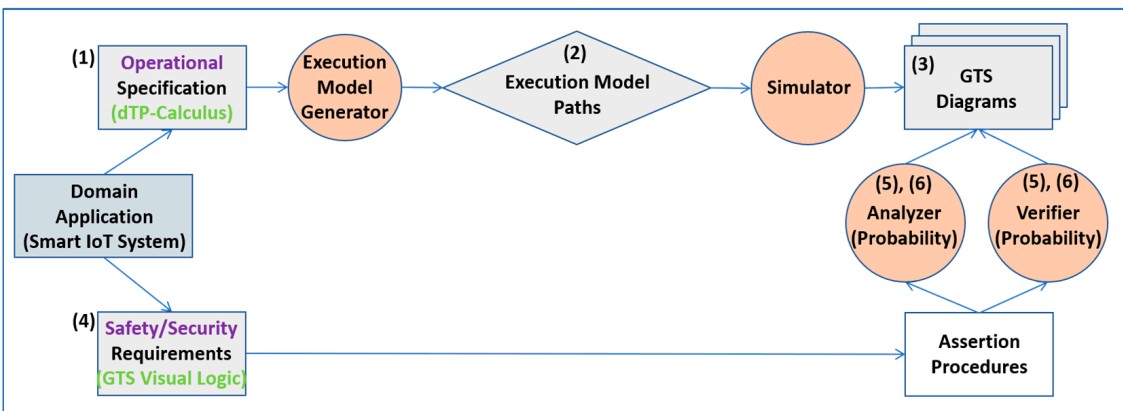

**Figure 1.** Overview of the approach.

Since the approach includes a number of models and methods, this paper will focus on the dTP-Calculus and its related models and present a brief description for GTS Logic and its related functionalities in the analysis and verification steps based on probability.

### 1.4. Proof of Concept

To demonstrate this approach, we developed the SAVE tool suite [18,19,21], on the ADOxx Meta-Modeling Platform from OMiLAB [22–24], to specify, analyze, and verify the basic and probabilistic requirements of Smart IoT Systems for Digital Twin using dTP-Calculus. It consists of System and Process Modeler, Execution Model Generator, Simulator, GTS Modeler, GTS Logic Specifier, Analyzer, and Verifier. As a proof of concept, a Smart EMS (Emergency Medical Service) system is selected to show the approach with static and dynamic probability of the choice operation for the example on the SAVE tool, and a miniature of Digital Twin for the example is constructed and described in the paper.

Note that OMiLAB Community provided more than 57 new methodologies, including tools, in the areas of education, IoT, business, etc., over a decade [22,23]. The methodologies perform domain-specific functionalities that extend the value of modeling over simple visual representations to support simple human understanding. The examples of the domains include haptic models [25], IoT simulation [18,19,21], robot controlling [26], etc. In the perspective of such domains, SAVE can be considered a methodological tool to model, analyze, and verify Digital Twin systems.

### 1.5. Contribution

The approach with the example on SAVE shows that dTP-Calculus is a very effective and efficient method to specify the nondeterministic behavior of the systems with probability, verify the safety and security requirements of the nondeterministic behavior, and control undesirable nondeterministic behavior. In general, it demonstrates how the unsatisfied requirements can be controlled by the given probabilities. In other words,

dTP-Calculus can be used to control the uncertainty of the requirements for Smart IoT Systems for Digital Twin with dynamic probability.

Further, the SAVE tool can provide the capability to implement the method for real industrial applications of Smart IoT Systems in Digital Twin. Most importantly, SAVE is available in the public domain as an open model in OMiLAB [27].

To demonstrate the feasibility of the approach for Digital Twin, a small-scale smart city was constructed for a SEMS system with smart mobile IoT devices as ambulances in the physical world. These devices are activated by the processes defined by dTP-Calculus in SAVE during the simulation in real-time in the virtual world.

### *1.6. Organization*

The paper is organized as follows: in Section 2, the syntax and semantics of dTP-Calculus will be described. In Section 3, the probabilistic specification and verification approach for the nondeterministic choice operation will be described. In Section 4, a Smart EMS (Emergency Medical Service) example for Digital Twin will be presented to demonstrate the feasibility of the approach on SAVE. In Section 5, a proof of concept will be presented for Digital Twin. In Section 6, a comparative analysis will be presented with other similar approaches in process algebra. Finally, conclusions and future research will be presented in Section 7.

## 2. dTP-Calculus

dTP-Calculus is a new process algebra designed by the authors. Note that dT-Calculus is the base of dTP-Calculus, designed by the authors too [18,19]. The basic characteristics of dT-Calculus are timed movement and geographical space of processes. The challenging characteristic of dTP-Calculus is the probabilistic choice operation with probability distributions.

### *2.1. Main Characteristics*

There are four main characteristics of dTP-Calculus: *mobility*, *synchronization*, *time*, and *probability*. The details of the main characteristics are as follows:

#### 2.1.1. Mobility

The main features of the movements in dTP-Calculus are *mode* and *direction* of movement:

(1)  Movement mode: Autonomy and heteronomy can be used to define the movement mode. The details for autonomy and heteronomy are as follows:

    ①    *Active* movements: A process moves into a target process voluntarily.
    ②    *Passive* movements: A target process is moved into other process involuntarily.

(2)  Movement direction: The direction of the movement can be classified as *inward* and *outward*.

    ①    *Move-in* direction: A process moves into its process autonomously.
    ②    *Move-out* direction: A process moves out of its immediate nesting process autonomously.
    ③    *Move-get* direction: A process is moved into a target process heteronomously by the target process.
    ④    *Move-put* direction: A process is moved out of its immediate nesting process heteronomously by the target process.

#### 2.1.2. Synchronization

The movements of dT-Calculus have a synchronous property. All the movements of processes in dT-Calculus require permission from the target processes, either the target process where a process moves into or out of in the autonomous case, or from a process that is moved into or out of by the target process in the heteronomous case. Asynchronous movement can be executed under limited conditions using the Priority property, as follows:

### 2.1.3. Priority

Priority is a property given to each process. Priority can be used as a condition for asynchronous movement. It also can be used to determine the execution order of processes when the processes are performed at the same time.

### 2.1.4. Time

Time can be used to specify the temporal constraints of processes in the systems. There are five types of temporal properties:

(1) *Ready Time*: The time to wait for the action before performing the action.
(2) *Timeout*: The maximum waiting time for the execution of the action. If the *Ready Time* has elapsed and the target process for its synchronous action is not prepared, the action cannot be performed.
(3) *Execution Time*: The time required for the action to be executed. The action is executed while the execution time is available. After the execution time is over, the action is terminated, and the next action is to be executed.
(4) *Deadline*: The time for terminating the execution of an action. All actions must be terminated by the Deadline. If the action is out of the Deadline, the process is in a fault state.
(5) *Period*: The duration of the action for recursion. The action repeats itself after executing the action during the specified period.

### 2.1.5. Probability

There are four types of probabilistic models [18,19] for dTP-Calculus:

(1) Discrete distribution without parameters.
(2) Normal distribution model with $\mu$ and $\sigma$.
(3) Exponential distribution model with $\lambda$.
(4) Uniform distribution based on the $u$ (upper bound) and $l$ (lower bound).

The discrete distribution is a probabilistic model that does not involve any parameters. It defines specific values of probabilities for each branch of the choice operation with some restrictions. One important constraint is that the probabilities assigned to all possible outcomes must add up to 100%. As a result, it can only be used to analyze the static probabilistic cases of specified systems, namely, Smart IoT Systems.

However, some variables can be included in the model to deal with dynamic probabilistic cases, as shown in this paper. As a result, it can be used to determine a threshold of the distribution at a satisfaction level for the safety and security requirements of Smart IoT systems. Detailed descriptions will be provided in the next subsections of the probabilistic choice operation.

### 2.2. Syntax

The details of the main characteristics are as follows: the dTP-calculus syntax is listed, as shown in Figure 2. Each construct of the dTP-Calculus syntax is described as follows:

(1) *Action* ($A$): Represents an action performed by a process, as shown in (S-1) of Figure 2. The types of actions are and Control (*Control*), Communication (*Send/Receive*), Movement (*Movement*), and Empty (*Null*).
(2) *Timed action* ($A^{per,n}_{[r,to,e,d]}$): Defines the execution of the action with temporal restrictions, as shown in (S-2) of Figure 2. It defines the process with timed properties. The temporal properties of [$r$, $to$, $e$, $d$] represent *Ready Time*, *Timeout*, *Execution Time*, and *Deadline*, respectively. *per* and $n$ mean a period of action and the number of repetitions, respectively.
(3) *Timed process* ($P^{per,n}_{[r,to,e,d]}$): Defines the process with timed properties, as shown in (S-3) of Figure 2.
(4) *Priority* ($P_{(pri\_n)}$): Defines the importance or urgency of the processes, as shown in (S-4) of Figure 2. *Priority* is expressed as an integer: a higher number implies higher

priority. Exceptionally, 0 indicates the highest priority, and an action with this priority can be used for asynchronous movement.

(5) *Nesting* ($P[Q]$): Defines that $P$ contains $Q$, as shown in (S-5) of Figure 2. Internal processes are controlled by external processes.

(6) *Channel* ($P\langle ch\rangle$): Represents a list of channels connected to processes, as shown in (S-6) of Figure 2, allowing the processes to communicate with other processes.

(7) *Choice* ($P + Q$): Defines that only one of several actions or one of several processes is selected for execution non-deterministically, as shown in (S-7) of Figure 2.

(8) *Probabilistic choice* ($P\{pc\} +_F Q\{pc\}$): Used to select one of the actions with a probability distribution, as shown in (S-8) of Figure 2. There are four probabilistic models as follows:

①  *Discrete Distribution* ($D$): Simply, probability is directly determined with discrete values, based on Discrete Distribution. For example, the processes $P$ and $Q$ are assigned with probabilities of 0.6 and 0.4, respectively. The syntax for probabilistic choice operations for Discrete Distribution is as follows:

$$P\{0.6\} +_D Q\{0.4\} \tag{1}$$

②  *Normal Distribution* ($N(\mu,\sigma)$): The parameters $\mu$ and $\sigma$ can be used to specify a desired Normal Distribution. For example, the values $\mu$ and $\sigma$ with the integers 40 and 10, respectively, can be used to specify the following probabilistic choice for Normal Distribution. The variable $v$ represents a probabilistic variable. For example, $v \leq 52$ indicates the probability that $v$ is less than or equal to 52 in a normal distribution with a mean $\mu$ of 40 and a standard deviation $\sigma$ of 10:

$$P\{v > 52\} +_{N(40,10)} Q\{v \leq 52\} \tag{2}$$

③  *Exponential Distribution* ($Ex(\lambda)$): The parameter $\lambda$ can be used to specify Exponential Distribution. For example, the value $\lambda$ with the real number 0.5 can be used to specify the following probabilistic choice for Exponential Distribution. The variable $v$ represents a probabilistic variable. For example, $v \leq 2.5$ indicates the probability that $v$ is less than or equal to 2.5 in an exponential distribution with a rate parameter $\lambda$ of 0.5:

$$P\{v > 2.5\} +_{E(0.5)} Q\{v \leq 2.5\} \tag{3}$$

④  *Uniform Distribution* ($U(l,u)$): The bound values lower ($l$) and upper ($u$) can be used to specify Uniform Distribution. For example, the values $l$ and $u$ with the integers 4 and 6, respectively, can be used to specify the following probabilistic choice for Uniform Distribution. The variable $v$ represents a probabilistic variable. For example, $v \leq 5$ indicates the probability that $v$ is less than or equal to 5 in a uniform distribution with a lower bound $l$ of 3 and an upper bound $u$ of 7:

$$P\{v > 5\} +_{U(3,7)} Q\{v \leq 5\} \tag{4}$$

(9) *Parallel* ($P \parallel Q$): Defines that two processes in a parallel relationship are executed at the same time, as shown in (S-9) of Figure 2.

(10) *Exception* ($P\backslash E$): Used to handle Deadline or Timeout, as shown in (S-10) of Figure 2.

(11) *Sequence* ($A \cdot P$): Used to specify the next action, as shown in (S-11) of Figure 2.

(12) *Empty* ($\phi$): An idle action, that is, *null*, as shown in (S-12) of Figure 2.

(13) *Send/Receive* ($ch(\overline{msg})/ch(msg)$): *Synchronous Communication* is based on describing the synchronous communication between processes, with two types of communication actions called *Send* and *Receive,* as shown in (S-13) and (S-14) of Figure 2.

(14) *Movement request* ($m^{pri}(k)P$): Defines a request for a process movement to another process, as shown in (S-15) of Figure 2.

(15) *Movement permission* ($Pm(k)$): Defines a permit for a process movement from other process, as shown in (S-16) of Figure 2.

(16) *Create process* (*newP*): Defines that a process creates a new internal process, as shown in (S-17) of Figure 2.

(17) *Kill process* (*killP*): Defines that a process terminates another process, as shown in (S-18) of Figure 2.

(18) *Exit process* (*exit*): Defines that a process terminates itself, as shown in (S-19) of Figure 2.

| | | | | | | |
|---|---|---|---|---|---|---|
| $P ::= A$ | Action | (S-1) | $A ::= \phi$ | Empty | (S-12) |
| $\mid A_{[r,to,e,d]}^{per,n}$ | Timed Action | (S-2) | $\mid ch(\overline{msg})$ | Send | (S-13) |
| $\mid P_{[r,to,e,d]}^{per,n}$ | Timed Process | (S-3) | $\mid ch(msg)$ | Receive | (S-14) |
| $\mid P_{(pri\_n)}$ | Priority | (S-4) | $\mid M$ | Movement | |
| $\mid P[Q]$ | Nesting | (S-5) | $\mid C$ | Control | |
| $\mid P\langle ch\rangle$ | Channel | (S-6) | $M ::= m^{pri}(k)\,P$ | Movement Request | (S-15) |
| $\mid P + Q$ | Choice | (S-7) | $\mid P\,m(k)$ | Movement Permission | (S-16) |
| $\mid P\{pc\} +_F Q\{pc\}$ | Probabilistic Choice | (S-8) | $m ::= in$ | In Movement | |
| $\mid P \parallel Q$ | Parallel | (S-9) | $\mid out$ | Out Movement | |
| $\mid P\backslash E$ | Exception | (S-10) | $\mid get$ | Get Movement | |
| $\mid A \cdot P$ | Sequence | (S-11) | $\mid put$ | Put Movement | |
| $F ::= D$ | Discrete Distribution | | $C ::= new\,P$ | Create Process | (S-17) |
| $\mid N(\mu,\sigma)$ | Normal Distribution | | $\mid kill\,P$ | Kill Process | (S-18) |
| $\mid Ex(\lambda)$ | Exponential Distribution | | $\mid exit$ | Exit Process | (S-19) |
| $\mid U(l,u)$ | Uniform Distribution | | | | |

**Figure 2.** Syntax of dTP-Calculus.

## 2.3. Semantics

Table 1 shows the semantics with a set of transition rules for dTP-Calculus. The transition rules define that a conclusion can be derived from a *Premise* when the *Side Condition* is satisfied:

$$Transition = \frac{Premise}{Conclusion}\,(Side\ Condition) \tag{5}$$

**Table 1.** Semantics of dTP-Calculus.

| No | Name | Transition Rules |
|---|---|---|
| (R-1) | *Sequence* | $\dfrac{-}{A\cdot P \xrightarrow{A} P}$ |
| (R-2) | *ChoiceL* | $\dfrac{-}{P+Q \rightarrow P}$ |
| | *ChoiceR* | $\dfrac{-}{P+Q \rightarrow Q}$ |
| (R-3) | *Probability Choice* | $\dfrac{A\cdot P \xrightarrow{A} P}{(\sum_{i\in I} A_i\{pc_i\})\cdot P \xrightarrow{A_i\{pc_i\}} P'.}\,(\sum_{i\in I} pc_i = 1,\ i \in I)$ |
| (R-4) | *Com* | $\dfrac{-}{ch_1(\overline{msg_1})\cdot P \parallel ch_2(msg_2)\cdot Q \xrightarrow{\tau} P \parallel Q}\,((ch_1 = ch_2) \wedge (msg_1 = msg_2))$ |
| (R-5) | *ParallelL* | $\dfrac{P \rightarrow P'}{P\parallel Q \rightarrow P'\parallel Q}$ |
| | *ParallelR* | $\dfrac{Q \rightarrow Q'}{P\parallel Q \rightarrow P\parallel Q'}$ |
| | *ParallelCom* | $\dfrac{P \xrightarrow{A} P',\,Q \xrightarrow{\overline{A}} Q'}{P\parallel Q \xrightarrow{\tau} P'\parallel Q'}$ |

**Table 1.** *Cont.*

| No | Name | Transition Rules |
|---|---|---|
| (R-6) | NestingO | $\dfrac{P \to P'}{P[Q] \to P'[Q]}$ |
| | NestingI | $\dfrac{Q \to Q'}{P[Q] \to P[Q']}$ |
| | NestingCom | $\dfrac{P \overset{A}{\to} P', Q \overset{\overline{A}}{\to} Q'}{P \,\|\, Q \overset{\tau}{\to} P' \,\|\, Q'}$ |
| (R-7) | In | $\dfrac{P \overset{in(k)Q}{\to} P', Q \overset{Pin(k)}{\to} Q'}{P \,\|\, Q \overset{\delta}{\to} Q'[P']}$ |
| | Out | $\dfrac{P \overset{out(k)Q}{\to} P', Q \overset{Pout(k)}{\to} Q'}{Q[P] \overset{\delta}{\to} P' \,\|\, Q'}$ |
| | Get | $\dfrac{P \overset{get(k)Q}{\to} P', Q \overset{Pget(k)}{\to} Q'}{P \,\|\, Q \overset{\delta}{\to} P'[Q']}$ |
| | Put | $\dfrac{P \overset{put(k)Q}{\to} P', Q \overset{Pput(k)}{\to} Q'}{P[Q] \overset{\delta}{\to} P' \,\|\, Q'}$ |
| (R-8) | InP | $\dfrac{P \overset{in^{pri}(k)Q}{\to} P'}{P_{(n_1)} \,\|\, Q_{(n_2)} \overset{\delta}{\to} Q_{(n_2)}[P'_{(n_1)}]} \left( (n_1 > n_2 \wedge n_2 \neq 0) \vee (n_1 = 0 \wedge n_2 \neq 0) \right)$ |
| | OutP | $\dfrac{P \overset{out^{pri}(k)Q}{\to} P'}{Q_{(n_2)}[P_{(n_1)}] \overset{\delta}{\to} P'_{(n_1)} \,\|\, Q_{(n_2)}} \left( (n_1 > n_2 \wedge n_2 \neq 0) \vee (n_1 = 0 \wedge n_2 \neq 0) \right)$ |
| | GetP | $\dfrac{P \overset{get^{pri}(k)Q}{\to} P'}{P_{(n_1)} \,\|\, Q_{(n_2)} \overset{\delta}{\to} P'_{(n_1)}[Q_{(n_2)}]} \left( (n_1 > n_2 \wedge n_2 \neq 0) \vee (n_1 = 0 \wedge n_2 \neq 0) \right)$ |
| | PutP | $\dfrac{P \overset{put^{pri}(k)Q}{\to} P'}{P_{(n_1)}[Q_{(n_2)}] \overset{\delta}{\to} P'_{(n_1)} \,\|\, Q_{(n_2)}} \left( (n_1 > n_2 \wedge n_2 \neq 0) \vee (n_1 = 0 \wedge n_2 \neq 0) \right)$ |
| (R-9) | TickTimeR | $\dfrac{-}{A^{per,n}_{[r,to,e,d]} \cdot P \overset{\rhd 1}{\to} A^{per,n}_{[r-1,to,e,d-1]} \cdot P} (r \geq 1)$ |
| (R-10) | TickTimeTO | $\dfrac{A \cdot P \,\|\, \overline{A} \cdot Q \overset{\tau \vee \delta}{\to} P \,\|\, Q}{A^{per,n}_{[0,to,e,d]} \cdot P \overset{\rhd 1}{\to} A^{per,n}_{[0,to-1,e,d-1]} \cdot P} (to \geq 1)$ |
| (R-11) | TickTimeSyncE | $\dfrac{A \cdot P \,\|\, \overline{A} \cdot Q \overset{\tau \vee \delta}{\to} P \,\|\, Q}{A^{per_1, \; n_1}_{[0,to_1,e_1,d_1]} \cdot P \,\|\, \overline{A}^{per_2, \; n_2}_{[0,to_2,e_2,d_2]} \cdot Q \overset{\rhd 1}{\to} A^{per_1, \; n_1}_{[0,to_1,e_1-1,d_1-1]} \cdot P \,\|\, \overline{A}^{per_2, \; n_2}_{[0,to_2,e_2-1,d_2-1]} \cdot Q} (e_1 \geq 1, e_2 \geq 1)$ |
| (R-12) | TickTimeAsyncE | $\dfrac{-}{A^{per,n}_{[0,to,e,d]} \cdot P \overset{\rhd 1}{\to} A^{per,n}_{[0,to,e-1,d-1]} \cdot P} (e \geq 1)$ |
| (R-13) | TickTimeEnd | $\dfrac{-}{A^{per,n}_{[0,to,0,d]} \cdot P \overset{\rhd 1}{\to} P}$ |
| (R-14) | Timeout | $\dfrac{-}{(A^{per,n}_{[0,0,e,d]} \backslash E) \cdot P \overset{\rhd 1}{\to} E \cdot P}$ |
| (R-15) | Deadline | $\dfrac{-}{(A^{per,n}_{[r,to,e,0]} \backslash E) \cdot P \overset{\rhd 1}{\to} E \cdot P}$ |
| (R-16) | Period | $\dfrac{-}{A^{per,n}_{[r,to,e,d]} \cdot P \overset{\rhd per}{\to} A^{per, \; n-1}_{[r,to,e,d]} \cdot P} (n \geq 1)$ |
| (R-17) | Period End | $\dfrac{-}{A^{per,0}_{[0,to,0,d]} \cdot P \overset{\rhd 1}{\to} P}$ |

The subsequent labeled transitions indicate that the process state $P$ can translate to a different process state $P'$, with or without Action $A$.

$$P \to P', \quad P \overset{A}{\to} P' \tag{6}$$

Each transition rule in the table is defined as follows:

(1)   *Sequence*: Indicates that if the action *A* is performed without the premise, the process *P* is executed with the action *A*, as shown in (R-1) of Table 1.

(2)   *Choice: ChoiceL* and *ChoiceR* represent the transition rules of the action. One of the two processes is executed under the premises while the other one is not executed, as shown in (R-2) of Table 1.

(3)   *Probability Choice*: Represents that the actions for choice are performed probabilistically with a given premise with a side condition. For example, $A_1\{0.7\} + A_2\{0.1\} + A_3\{0.2\}$ implies that the probabilities for Actions $A_1$, $A_2$, $A_3$ are 70%, 10%, 20%, respectively, as shown in (R-3) of Table 1.

(4)   *Com*: Defines the synchronous communication between *P* and *Q* on a channel with the conditions of $ch_1 = ch_2$ and $msg_1 = msg_2$. The *Send* action is defined by a message with an overline ($\overline{msg1}$) and the *Receive* action is defined by a message without an overline ($msg_2$). Synchronous communication is represented by the $\tau$ action, as shown in (R-4) of Table 1.

(5)   *Parallel: ParallelL* and *ParallelR* mean that the processes *P* and *Q* are in parallel and executed independently. However, in the case of dependency, it is necessary to apply the *ParallelCom* rule. If two processes *P* and *Q* are synchronous, their $\tau$ action can be executed synchronously in parallel, without causing any impact on other processes, as shown in (R-5) of Table 1.

(6)   *Nesting: NestingO* and *NestingI* mean that the processes *P* and *Q* can be performed independently without synchronization. However, if they are synchronous, the parallel synchronous transition as *NestingCom* will have an impact on both processes. It is important that the synchronous action between *P* and *Q* is represented by the $\tau$ action, as shown in (R-6) of Table 1.

(7)   *In*, *Out*, *Get*, *Put: In* and *Get* mean that a process moves into its target process, autonomously and heteronomously, respectively. *Out* and *Put* mean that a process moves out of its nesting process autonomously and heteronomously, respectively. The movements of dTP-Calculus are synchronous. All the movements of processes in dT-Calculus require permission from the target processes. Such synchronous movement actions are represented by the $\delta$ action. Note that *In* and *Out* are active, and *Get* and *Put* are passive, as shown in (R-7) of Table 1.

(8)   *InP*, *OutP*, *GetP*, *PutP:* These rules indicate the asynchronous movements between two processes. The asynchronous movements can be decided by priorities between two processes. If the priority of the process to request the target process is higher than the target process, the permission of the target process is not required to move to another space. These rules can be used to handle some exceptional cases in emergency situations, as shown in (R-8) of Table 1.

(9)   *TickTimeR:* Transition rule for Ready Time of the action by decrementing the ready time *r* and the deadline *d* of an action by 1 unit time with $\triangleright 1$, as shown in (R-9) of Table 1. Note that $\triangleright 1$ implies the elapse of 1 unit time.

(10)   *TickTimeTO:* Transition rule for the waiting time of the action by decrementing the timeout *to* and the deadline *d* of an action by a time unit with $\triangleright 1$ after the ready time *r* is completed in a condition that the synchronous partner process is not ready, as shown in (R-10) of Table 1. As stated, $\triangleright 1$ implies the elapse of 1 unit time.

(11)   *TickTimeSyncE:* Transition rule for the Execution Time of the synchronous action. When both actions *A* and $\overline{A}$ are ready simultaneously, they are executed synchronously, and the execution time *e* and the deadline *d* of the actions are decremented by a time unit with $\triangleright 1$, as shown in (R-11) of Table 1. As stated, $\triangleright 1$ implies the elapse of 1 unit time.

(12)   *TickTimeAsyncE:* Transition rule for the asynchronous action. Since the asynchronous action does not require waiting for its timeout *to*, it is possible to proceed to its execution just after its ready time *r*. After that, its execution time *e* and deadline *d* are decremented by a time unit with $\triangleright 1$, as shown in (R-12) of Table 1. As stated, $\triangleright 1$ implies the elapse of 1 unit time.

(13) *TickTimeEnd:* Defines the termination of the action *A* by completing its execution time *e*, as shown in (R-13) of Table 1.

(14) *Timeout:* Indicates the transition rule for the occurrence of timeout when *Timeout*(*to*) becomes 0 by the elapse of a time unit with $\triangleright$ 1, which implies a system fault, as shown in (R-14) of Table 1. As stated, $\triangleright$ 1 implies the elapse of 1 unit time. If an exception handler *E* is defined and the action with the fault is terminated, the handler *E* after the exception operator (\\) is executed. Note that Process *P* is still valid.

(15) *Deadline:* Indicates the transition rule for the violation of the deadline, as shown in (R-15) of Table 1. *Deadline*(*d*) becomes 0 by the elapse of a time unit with $\triangleright$ 1, which implies a system fault. As stated, $\triangleright$ 1 implies the elapse of 1 unit time. If an exception handler *E* is defined, the action with the fault is terminated, and the handler process *E* after the exception operator (\\) is executed. Process *P* is still valid.

(16) *Period:* Defines the rule for the execution of a periodic action *A*, as shown in (R-16) of Table 1. In *Period*, Action *A* executes itself in *n* times. This means that the value of *n* will be decremented by 1 after each $\triangleright$ *per*. As stated, $\triangleright$ *per* implies the elapse of 1 unit time.

(17) *Period End:* Defines the rule for termination of the periodic action *A*, as shown in (R-17) of Table 1. Since the value of *n* is 0, Action *A* will not repeat itself anymore after the elapse of a time unit, $\triangleright$ 1. As stated, $\triangleright$ 1 implies the elapse of 1 unit time.

## 3. Conceptual Approach

This section presents a conceptual overview of the approach in the paper with a simple example, based on the following specification, analysis, and verification steps.

### 3.1. Specification Step

Figure 3 shows a simple *PBC* (*Producer-Buffer-Consumer*) example in dTP-Calculus. It consists of *Producer* (*P*), *Buffer* (*B*), and *Consumer* (*C*). Note that the *Producer* produces two resources: *Resource*1 (*R1*) and *Resource2* (*R2*). The operational requirements with probability for *PBC* are as follows:

(1) *P* has the resources: *R1* and *R2*.

(2) *P* puts the resources in *B* in order: *R1-R2* or *R2-R1*.

(3) *P* sends a signal for the sending order for the resources to *B*.

    ① The probability of the *R1-R2* order for *P* is 0.6; that of its reverse, *R2-R1*, is 0.4.

    ② The probability of the *R1-R2* order for *B* is 0.7; that of its reverse, *R2-R1*, is 0.3.

(4) The resources from *B* are handled by *C* in that order from the above (3).

(5) *C* sends a signal of the order to *B*.

    ① The probability of the *R1-R2* order for *C* is 0.8; that of its reverse, *R2-R1*, is 0.2.

    ② The probability of the *R1-R2* order for *B* is 0.5; that of its reverse, *R2-R1*, is 0.5.

$$PBC = P[R1 \parallel R2] \parallel B \parallel C;$$
$$P = \left(PB(\overline{SendR1})\{0.6\}.\,putR1.\,putR2 +_D PB(\overline{SendR2})\{0.4\}.\,putR2.\,putR1\right).\,exit;$$
$$B = \left(PB(SendR1)\{0.7\}.\,getR1.\,getR2 +_D PB(SendR2)\{0.3\}.\,getR2.\,getR1\right).$$
$$\left(CB(SendR1)\{0.5\}.\,putR1.\,putR2 +_D CB(SendR2)\{0.5\}.\,putR2.\,putR1\right).exit;$$
$$C = \left(CB(\overline{SendR1})\{0.8\}.\,getR1.\,getR2 +_D CB(\overline{SendR2})\{0.2\}.\,getR2.\,getR1\right).\,exit;$$
$$R1 = P\,put.\,B\,get.\,B\,put.\,C\,get.\,exit;$$
$$R2 = P\,put.\,B\,get.\,B\,put.\,C\,get.\,exit;$$

**Figure 3.** Syntax of dTP-Calculus for the PBC example.

Figure 4 shows the pictorial view of *PBC*. The large circles indicate the processes *P*, *B*, and *C*. The small circles in the process *P* indicate the child processes *R*1 and *R*2. Note that There are *PB* and *CB* channels between *P* and *B*, and *C* and *B*, respectively, to exchange messages with each other. The unconditional choices are selected non-deterministically when the choice operations are performed on the channels. There are four possible synchronous combinations with probabilities on each channel, that is, *PB* and *CB*: (1) those with 0.42, 0.28, 0.18, and 0.12 probabilities from the 0.6 vs. 0.4 of *P* by the 0.7 vs. 0.3 of *B*, (2) those with 0.40, 0.40, 0.10, and 0.10 probabilities from the 0.5 vs. 0.5 of *B* by the 0.8 vs. 0.2 of *C*. The rules *Parcom* and *ProbabilityChoice* in Table 1 can be used as the condition for four possible synchronous combinations.

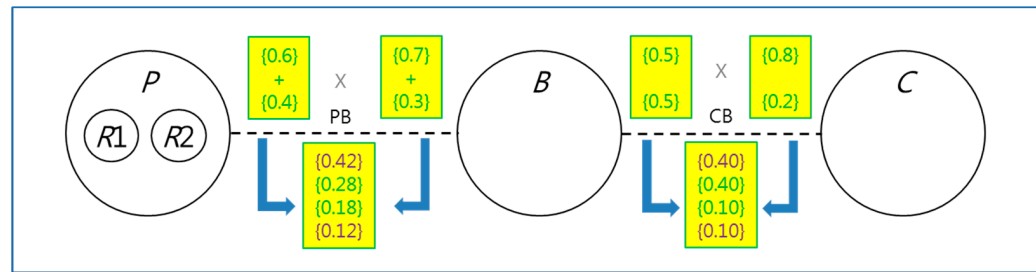

**Figure 4.** System View for the PBC example with probabilities.

*3.2. Analysis Step*

Figure 5 shows the reachability graph of the *PBC* example in the perspective of probabilistic choice operation. The top node is the root node, that is, the starting point of the execution. Synchronous communication can be derived from the interaction between the processes *P* and *B*. As a result, four possible synchronous combinations are generated from the interactions between the processes *P* and *B* on the first channel, *PB*, as shown on the left side of Figure 4, with the 0.42, 0.28, 0.18, and 0.12 probabilities, calculated from the 0.6 vs. 0.4 of *P* by the 0.8 vs. 0.2 of *B*. Note that there are two deadlock cases in the middle. The normal cases are only two cases, that is, the left-most and the right-most, from which another series of combinations of the following synchronous communication on *CB* between *B* and *C* can be extended. These are the same ones with those cases on the second channel, *CB*, as shown on the right side of Figure 4, with the 0.40, 0.40, 0.10, and 0.10 probabilities, calculated from the 0.8 vs. 0.2 of *C* by the 0.5 vs. 0.5 of *B*. However, in Figure 5, the normal cases are affected by the previous combination: (1) those of 0.168, 0.168, 0042, and 0.042 probabilities for the right-most, and (2) those of 0.048, 0.048, 0.012, and 0.012 probabilities for the left-most. As with the previous combinations, there are two deadlock cases for each combination in the middle. The total 0.27 probability for the normal cases are generated at the node at the bottom of Figure 5.

*3.3. Verification Step*

For the *PBC* example, the following types of requirements are specified:

(1)  Security Requirements:

   ①   *ScReq*1: The order of *R1*-*R2* or *R2*-*R1* should not be violated because the security information is contained in the first resource to decode the second resource.

   ②   *ScReq*2: The time interval between the first and second should not exceed 3 time units.

(2)  Safety Requirement:

   ①   *SfReq*1: *C* should consume the resources, which are produced by *P*, in less than 10 time units.

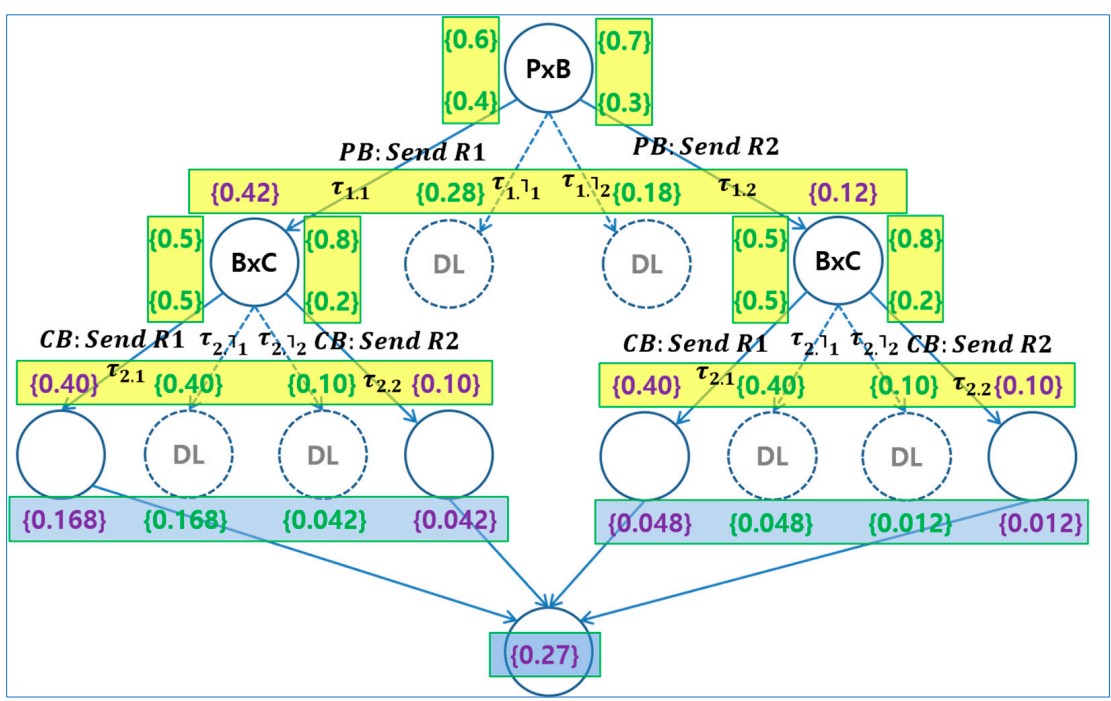

**Figure 5.** Probabilistic reachability graph of the PBC Example.

In addition, the security and safety requirements are defined from different perspectives. The security requirement is aimed at preventing an accident caused by external problems, while the safety requirement is aimed at preventing an accident caused by internal problems.

Table 2 shows the analysis of probabilistic verifications of these requirements. The values in Table 2 are derived from the probabilistic execution model for *PBC*.

**Table 2.** Analysis of probabilistic verification for the requirements.

| | eP1 | eP2 | eP3 | eP4 | eP5 | eP6 | eP7 | eP8 | eP9 | eP1 | **Total** |
|---|---|---|---|---|---|---|---|---|---|---|---|
| $\tau_1 \cdot \tau_2$ | $\tau_{1,1}\tau_{2.1}$ | $\tau_{1,1}\tau_{2.\daleth1}$ | $\tau_{1,1}\tau_{2.\daleth2}$ | $\tau_{1,1}\tau_{2.2}$ | $\tau_{1,\daleth1}$ | $\tau_{1,\daleth2}$ | $\tau_{1,2}\tau_{2.1}$ | $\tau_{1,2}\tau_{2.\daleth1}$ | $\tau_{1,2}\tau_{2.\daleth2}$ | $\tau_{1,2}\tau_{2.2}$ | |
| Prob. | 0.168 | 0.168 | 0.042 | 0.042 | 0.28 | 0.28 | 0.048 | 0.048 | 0.012 | 0.012 | 1.00 |
| *ScReq*1 | ○ | × | × | × | × | × | × | × | × | ○ | 0.18 |
| *ScReq*2 | ○ | × | × | ○ | × | × | ○ | × | × | ○ | 0.27 |
| *SfReq*1 | ○ | × | × | ○ | × | × | ○ | × | × | ○ | 0.27 |

There are execution paths in Table 2. Each path is represented by the type of communication. The details for $\tau_1$ and $\tau_2$ for the communication are as follows:

①   $\tau_{1,1}$: The order of *R1-R2* with the communication between the processes *P* and *B*, represented by the execution paths *eP*1 and *eP*4 in Table 2.

②   $\tau_{1,2}$: The order of *R2-R1* with the communication between the processes *P* and *B*, represented by *eP*7 and *eP*10 in Table 2.

③   $\tau_{2,1}$: The order of *R1-R2* with the communication between the processes *B* and *C*, represented by *eP*4 in Table 2.

④   $\tau_{2,2}$: The order of *R2-R1* with the communication between the processes *B* and *C*, represented by *eP*10 in Table 2.

Additionally, the following list represents the failures of the communications:

①   $\tau_{1,\daleth1}$: Failure of $\tau_{1,1}$, by *eP*5 in Table 2.

②   $\tau_{1,\daleth2}$: Failure of $\tau_{1,2}$, by *eP*6 in Table 2.

③     $\tau_{1,1} \cdot \tau_{2,\tau_1}$: Failure of $\tau_{2,1}$ after $\tau_{1,1}$, by *eP2* in Table 2.

④     $\tau_{1,1} \cdot \tau_{2,\tau_2}$: Failure of $\tau_{2,2}$ after $\tau_{1,1}$, by *eP3* in the Table 2.

⑤     $\tau_{1,2} \cdot \tau_{2,\tau_1}$: Failure of $\tau_{2,1}$ after $\tau_{1,2}$, by *eP8* in Table 2.

⑥     $\tau_{1,1} \cdot \tau_{2,\tau_2}$: Failure of $\tau_{2,2}$ after $\tau_{1,2}$, by *eP9* in Table 2.

First, for the purpose of verifying *ScReq1*, the order of *R1-R2* or *R2-R1* should be checked as follows:

①     $\tau_{1,1} \cdot \tau_{2,1}$: The order of *R1-R2*, by *eP1* in Table 2.

②     $\tau_{1,2} \cdot \tau_{2,2}$: The order of *R2-R1*, by *eP10* in Table 2.

For the requirements *SfReq1* and *ScReq2*, note that all the actions and interactions consume only 1 time unit. Consequently, the total delivery time of the resources takes 7 time units for both *R1-R2* and *R2-R1*.

Further, it can be seen in Table 2 that the probabilities for the satisfaction of the requirements *ScReq1*, *ScReq2*, and *SfReq1* are 0.18, 0.27, and 0.27, respectively.

## 4. Smart EMS Example Using SAVE

The purpose of this section is to prove the applicability of dTP-Calculus for a Smart IoT System, namely *Smart EMS* (*Emergency Medical Service*) (*SEMS*) in Digital Twin, using SAVE. Note that SAVE is a tool developed on the ADOxx Meta-Modeling Platform to prove the basic concept of the approach in this paper.

### 4.1. ADOxx Meta-Modeling Platform

ADOxx is a meta-modeling platform developed by OMiLAB [22–24]. It provides the facilities to develop modeling tools in various business and engineering domains. Figure 6 shows the basic components of the meta-modeling facilities of ADOxx as follows:

(1) Modeling Language: Provides a set of facilities and libraries to define notation, syntax, and semantics of modeling languages from different domains.

(2) Modeling Technique: Provides a set of facilities and libraries to define procedures to construct models in the modeling languages from above (1) and obtain the expected results from the models.

(3) Mechanisms and Algorithms: Provides a set of facilities and libraries to utilize the basic and enhanced functionalities of generic mechanisms and their algorithms in the meta-modeling platform for developing new modeling methods in common domains.

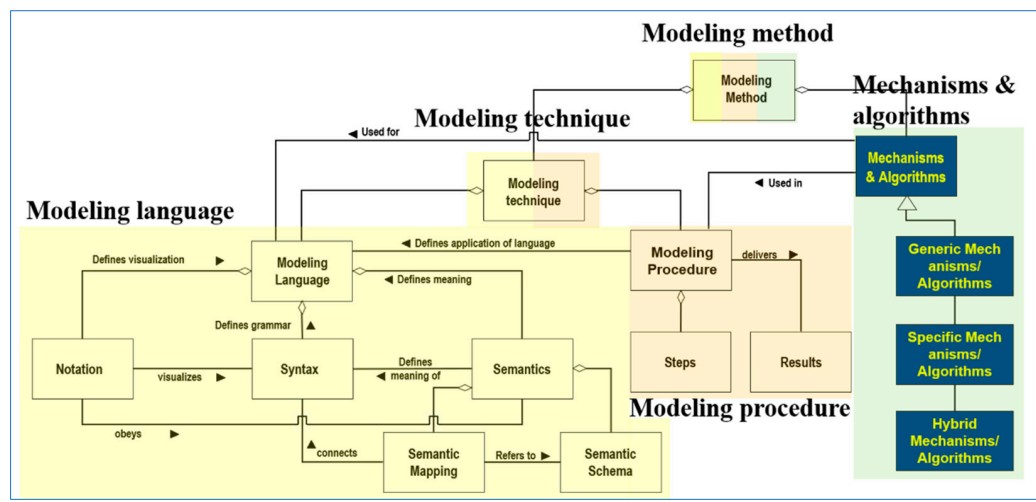

**Figure 6.** The basic components of ADOxx [22,23].

Figure 7 shows the basic facilities and libraries of ADOxx for developing a specific modeling tool.

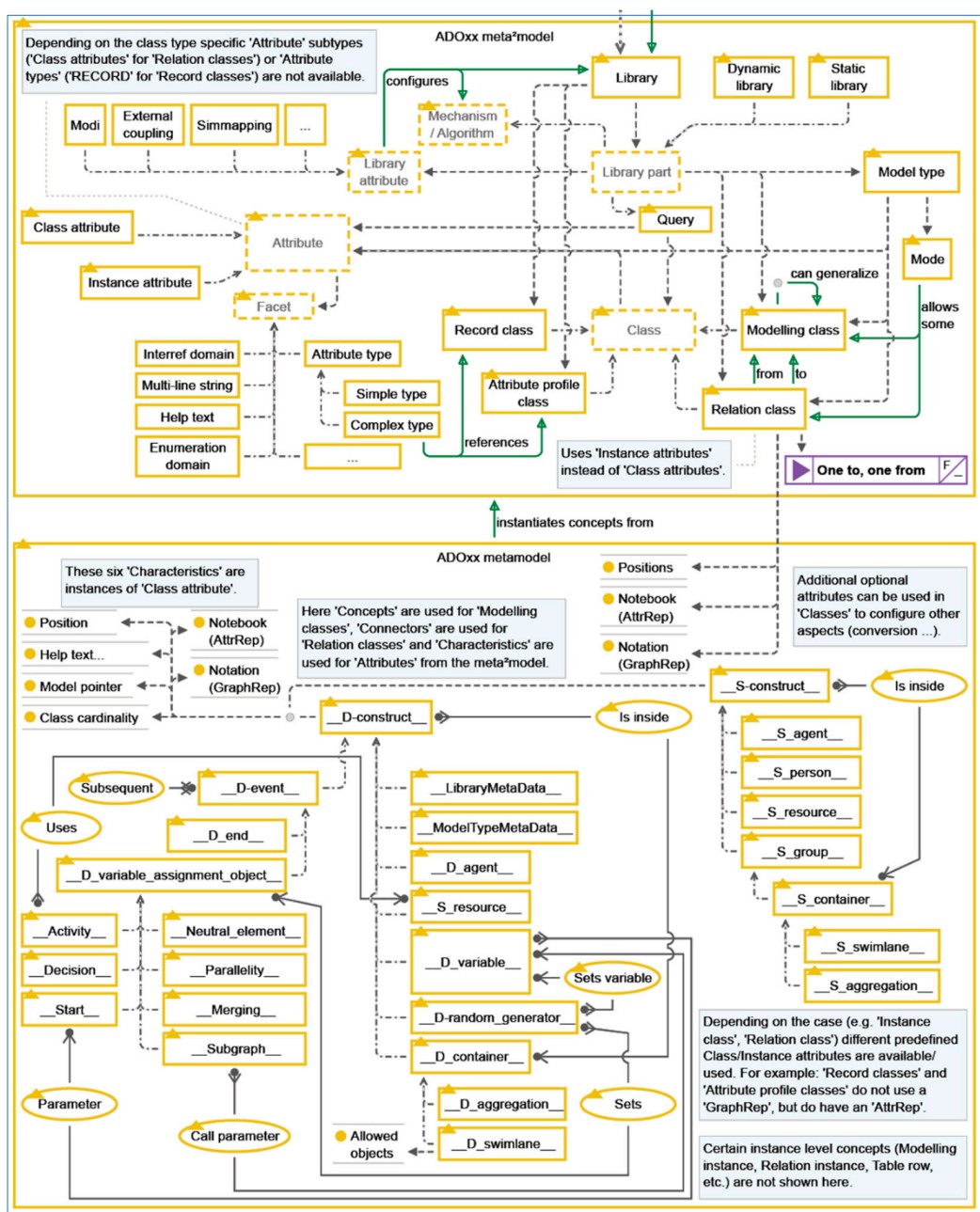

**Figure 7.** ADOxx Basic Facilities and Meta² Model Concepts: Design with CoChaCo [28].

### 4.2. SAVE

SAVE (Specification, Analysis, Verification, and Evaluation) [18,19,21] is a tool suite for dTP-Calculus, developed on the ADOxx Meta-Modeling Platform [22,23], as shown in Figure 8. The components of the SAVE tool are the Specifier, Analyzer, and Verifier. There are engines which consist of sub-engines in the components of the SAVE Tool.

The meta-modeling definition of the ITL (In-The-Large) and ITS (In-The-Small) views is visually specified in Tables 3 and 4, and their notations are shown at the bottom of each table. The detailed definition and representations are described in [18,19,21]. The detailed descriptions of the engines of the Specifier are as follows:

(1)   ITL/ITS Loader: In-The-Large (ITL) is the model that shows the inclusion relations of the processes in the systems. In-The-Small (ITS) is the model that shows the detailed actions of the processes of the systems. ITL/ITS Loader loads all the information from an ITL and its included ITS's.

(2) ITL/ITS Mapper: ITL/ITS Mapper is the syntax checker for the pair of an ITL and its included ITS's, and it creates the preliminary data for the EM Generator.

(3) dTP-Cal. Syntax Checker: dTP Cal. Syntax Checker checks whether the syntax of dTP-Calculus is correct or not when T2M(Text2Model) Parser is running.

(4) T2M Parser: T2M(Text2Model) Parser parses the textual specification with dTP-Calculus to change it into an ITL and its included ITS's.

(5) ITL/ITS Model Generator: ITL/ITS Model Generator generates the pair of an ITL and its included ITS's at the end of the T2M Parser process.

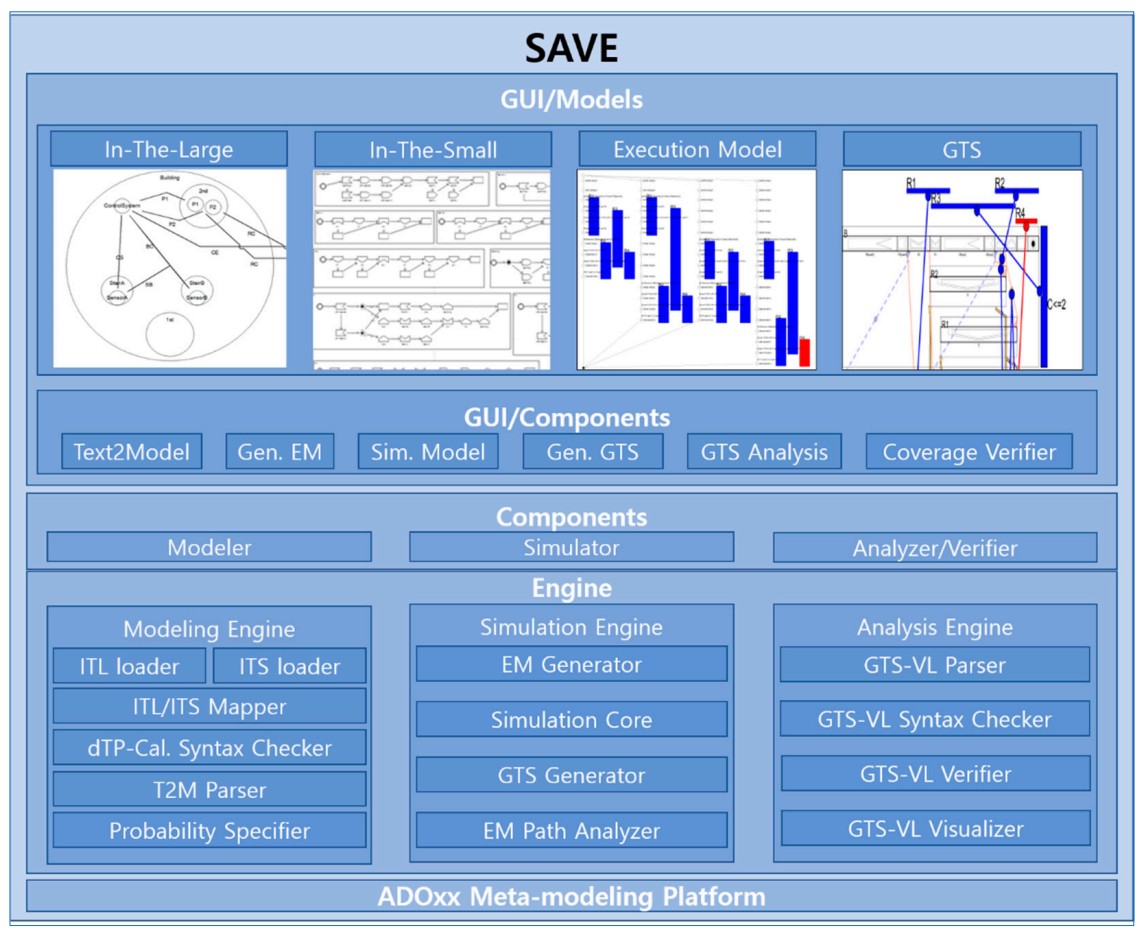

**Figure 8.** SAVE Architecture.

**Table 3.** Meta-Model Definition for System Model.

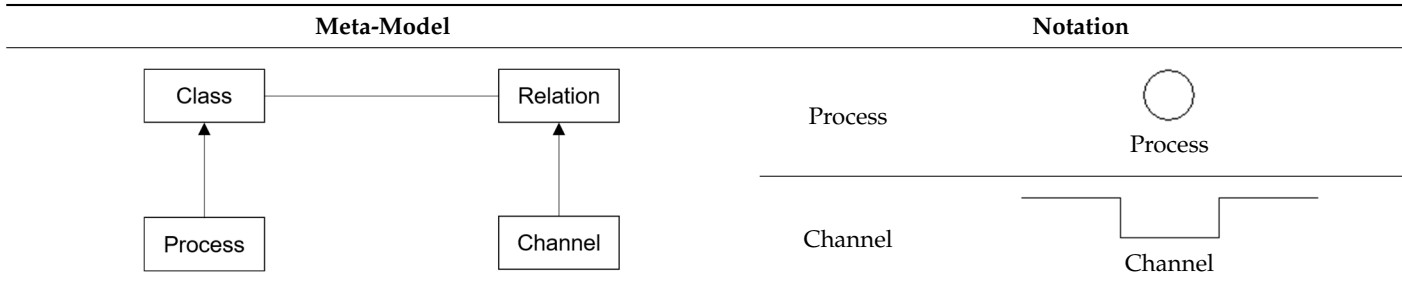

**Table 4.** Meta-Model Definition for Process Model.

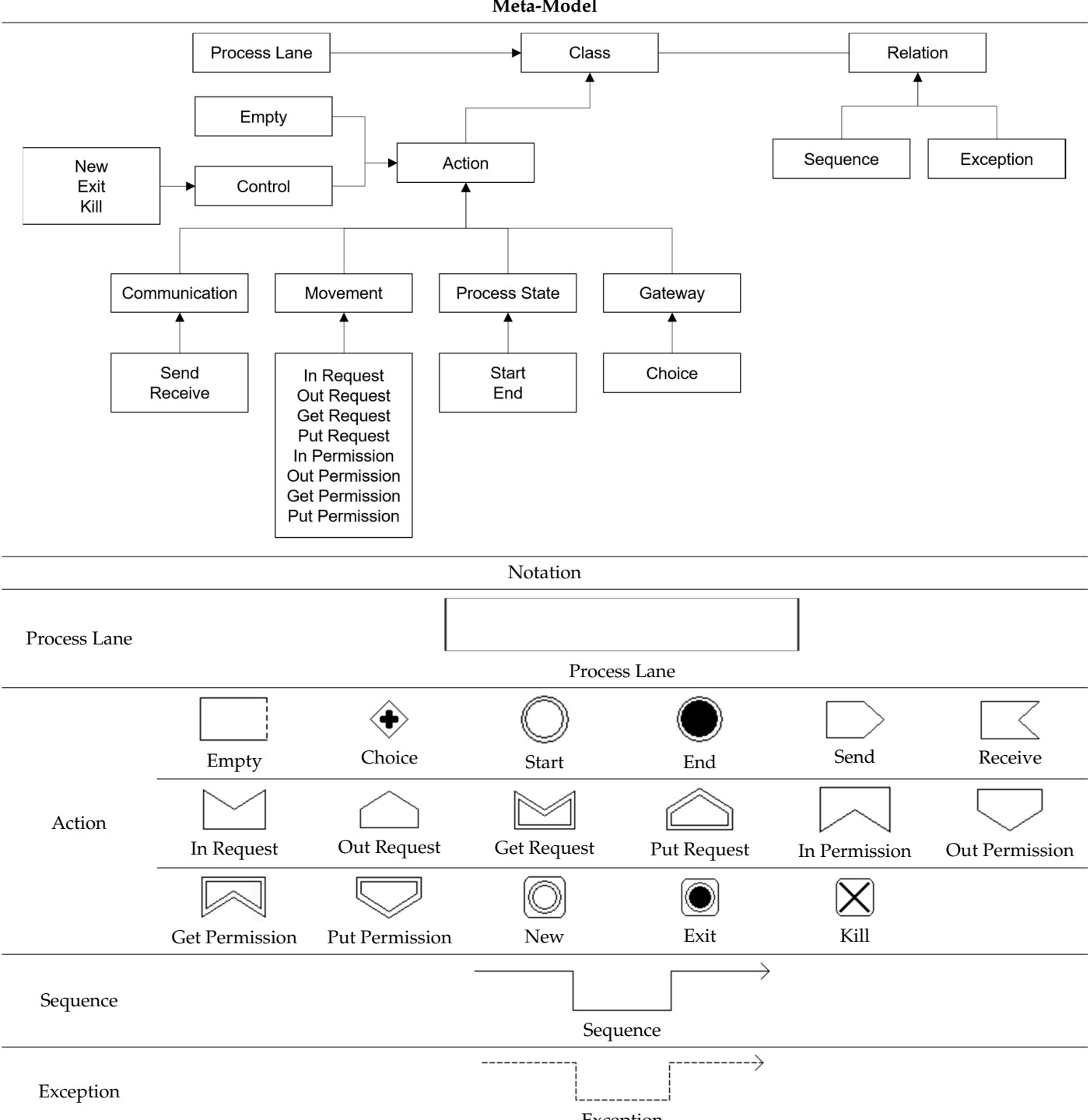

The meta-modeling definition of the execution model is visually specified in Table 5, and its notations are shown at the bottom of the table. The detailed definition and representations are described in [18,19,21]. The detailed descriptions of the engines of the simulator are as follows:

(1) EM Generator: EM Generator generates the execution model for the pair of an ITL and its included ITS's. The execution model is the state transition model, which can

be used to identify all the possible execution paths of the systems and display all their transitions in the tree form.

(2) GTS Generator: GTS Generator generates GTS. GTS is the diagram that shows all the actions of the processes of the systems with the blocks in the 2-dimensional space.

(3) EM Path Analyzer: EM Path Analyzer analyzes all the execution paths in the execution model.

**Table 5.** Meta-Model Definition for Execution Model.

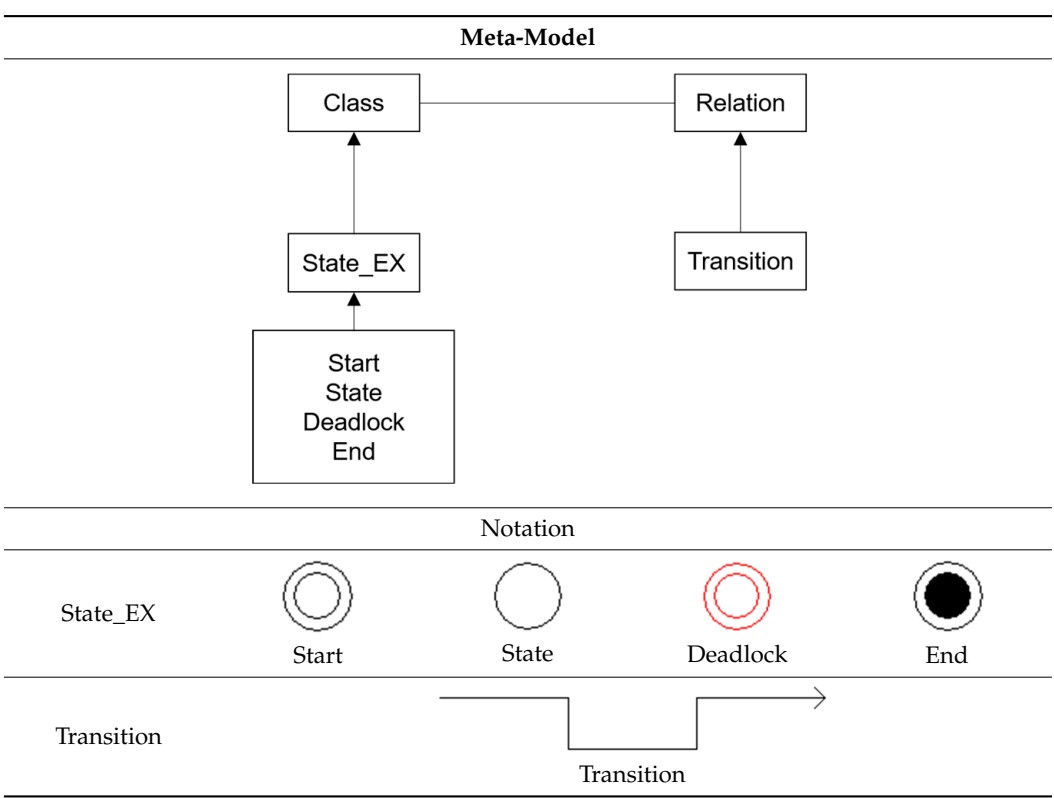

The meta-modeling definition of the GTS model is visually specified in Table 6, and its notations are shown at the bottom of the table. The detailed definition and representations are described in [18,19,21]. The detailed descriptions of the engines of the analyzer and the verifier are as follows:

(1) GTS-VL Parser: GTS-VL Parser parses the safety requirements of the systems with GTS-VL to verify the safety requirements of the systems.

(2) GTS-VL Syntax Checker: GTS-VL Syntax Checker checks whether the syntax of GTS-VL is correct or not when GTS-VL Parser is running.

(3) GTS-VL Verifier: GTS-VL Verifier analyzes and verifies the safety requirements of the systems.

(4) GTS-VL Visualizer: GTS-VL Visualizer visualizes the verification results of the safety requirements of the systems.

**Table 6.** Meta-Model Definition for GTS Model.

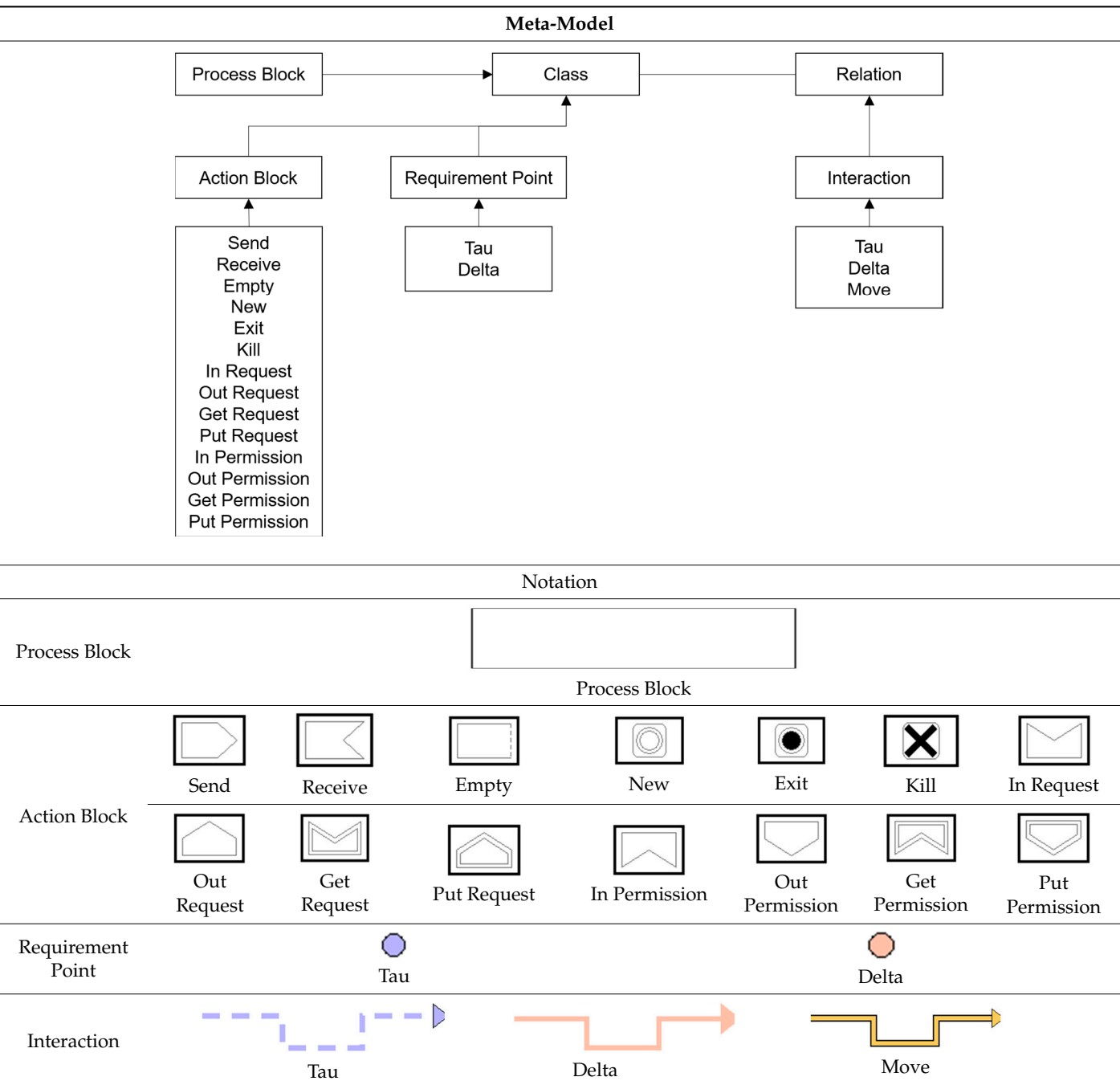

*4.3. Example*

4.3.1. Description

Figure 9 shows the code for the *SEMS* example with dTP-Calculus syntax. Note that the figure is also included in Appendix A for clearer visualization. It includes *911 Center*, *Patient*s, *Place*s, *Ambulance*s, and *Hospital*s. Their inclusion relations and geographical distribution can be represented in the ITL (System) view, as shown in Figure 10. The definitions of the processes, namely ITS (Process, omitted), specify the detailed actions for the processes. The main goal of the system is to transport four types of patients to two designated hospitals according to the types of urgency and illness, such as food poisoning (FP), high blood pressure (HBP), and heart disease (HD). Two ambulances are responsible for transporting the patients as informed by *911 Center* within the deadlines.

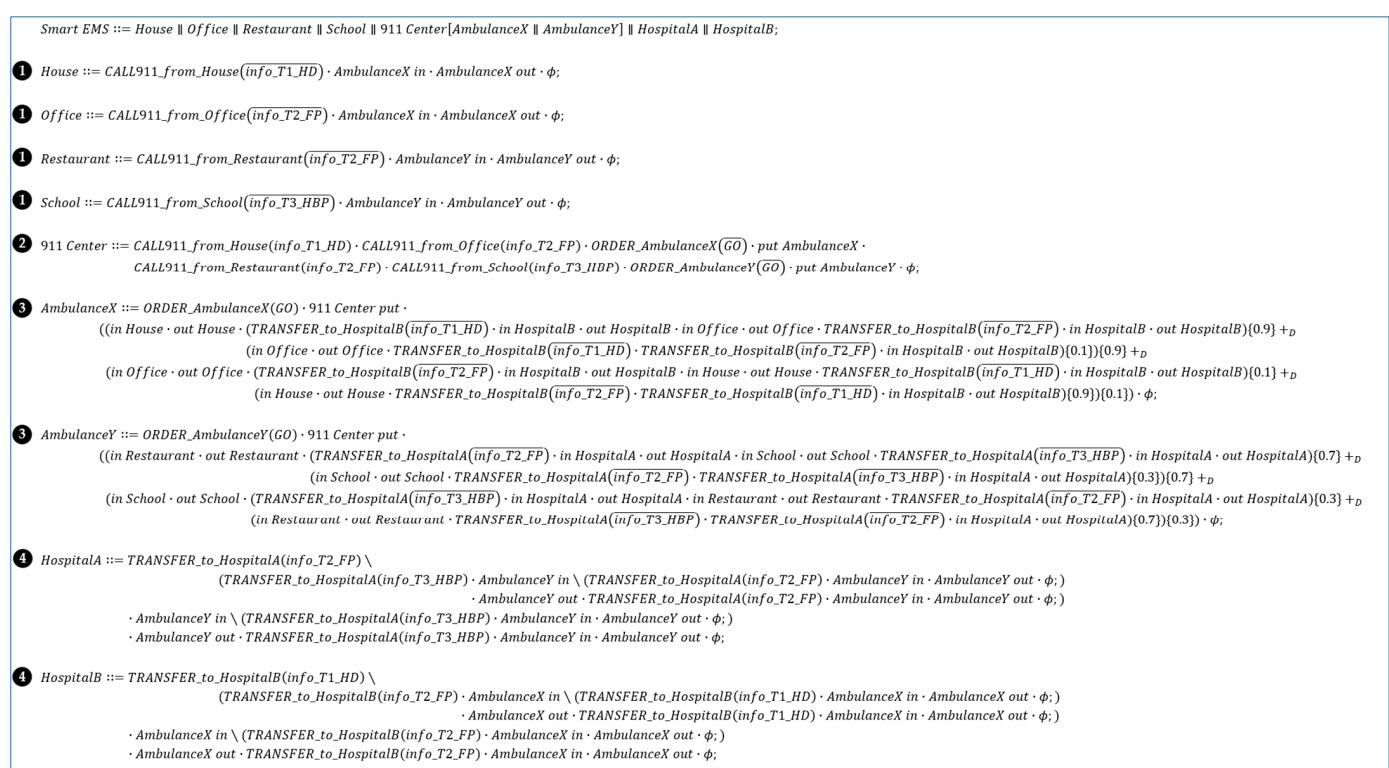

**Figure 9.** SEMS Specification in dTP-Calculus.

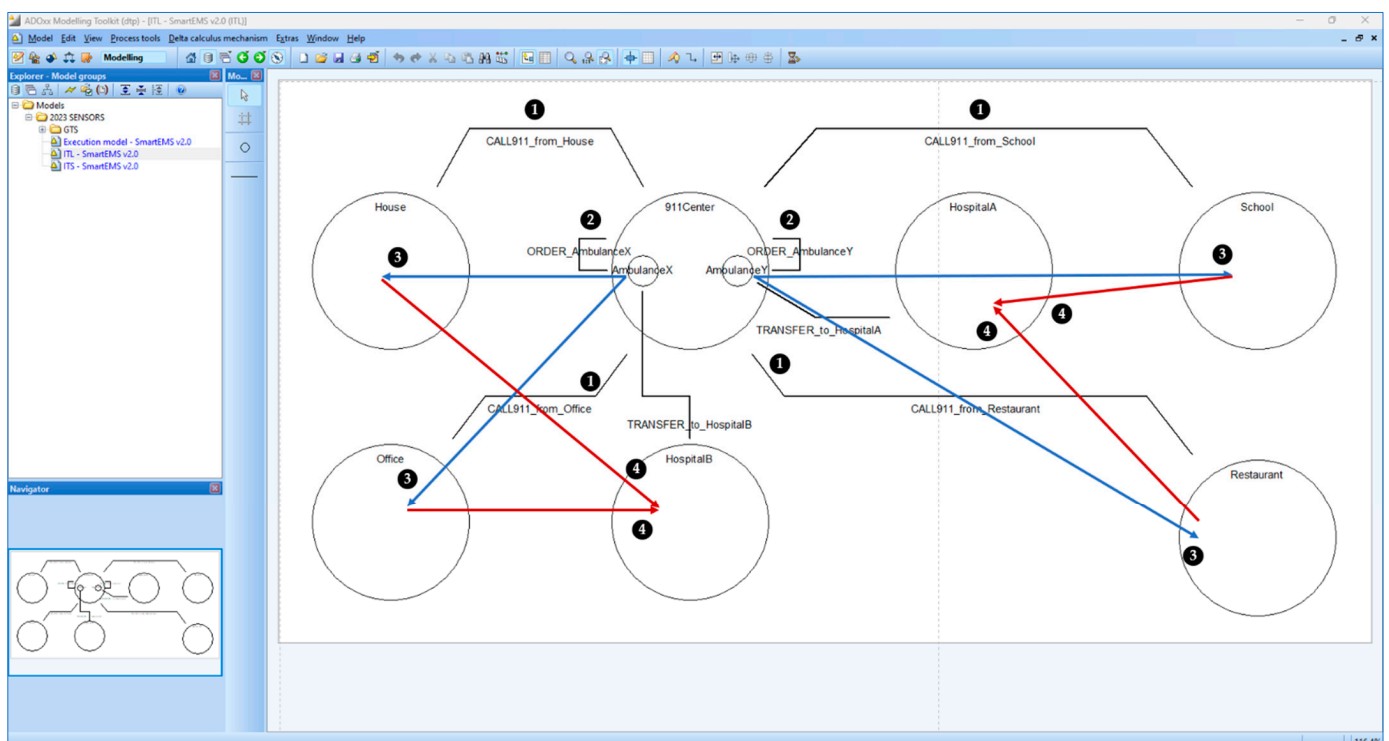

**Figure 10.** ITL View for SEMS.

### 4.3.2. Specification

Figure 9 shows the dTP-Calculus code, and Figure 10 represents the dTP-Calculus code in the ITL view for *SEMS*, generated by the SAVE tool. The processes in *SEMS* are defined as follows:

(1) *911 Center*: Patients send emergency calls to *911 Center* from their locations. *AmbulanceX* and *AmbulanceY* receive the emergency calls and orders from *911 Center* to transport the patients to the hospitals. Table 7 shows the priority information for the patients. Note that *AmbulanceX* and *AmbulanceY* are in *911 Center*.
(2) *Ambulance*: Two ambulances, specifically designated as *AmbulanceX* and *AmbulanceY*.
(3) *Place*: Four distinct locations, namely *School*, *Restaurant*, *Office*, and *House*, where the patients are dispersed.
(4) *Hospital*: Two medical facilities, namely *HospitalA* and *HospitalB*.

**Table 7.** Priorities for patients.

| Triage | | Priority | Illness |
|---|---|---|---|
| T1 | Immediate | Life-threatening patient (1st Priority) | *HD* |
| T2 | Delayed | Operation-needed Patient (2nd Priority) | *FP* |
| T3 | Minimal | Examination-needed Patient (3rd Priority) | *HBP* |

The following description outlines the scenario of *SEMS*. Note that each action is numbered in Figures 9 and 10, with the circled numbers indicating the corresponding processes in the dTP-Calculus code in Figure 9 and their graphical representations in Figure 10, respectively:

(1) *911 Center* receives the emergency calls from the patients in *House* and *Office*: ①.

  (i) A *T1* patient in *House*.
  (ii) A *T2* patient in *Office*.

(2) *House* and *Office* send *911 Center* information about the patients, and *AmbulanceX* receives the information from *911 Center*: ②.
(3) *911 Center* receives emergency calls from the patients in *Restaurant* and *School*: ①.

  (i) A *T2* patient in *Restaurant*.
  (ii) A *T3* patient in *School*.

(4) *Restaurant* and *School* send *911 Center* information about the patients, and *AmbulanceY* receives the information from *911 Center*: ②.
(5) *AmbulanceX* moves to *Office* and *House*, and the patients are transported by *AmbulanceX* to *HospitalB*: ③.
(6) *AmbulanceY* moves to *School* and *Restaurant*, and the patients are transported by *AmbulanceY* to *HospitalA*: ③.
(7) The patients from *School* and *Restaurant* are provided with medical treatment in *HospitalA*: ④.
(8) The patients from *Office* and *House* are provided with medical treatment in *HospitalB*: ④.

### 4.3.3. Probability Analysis

Figure 11 shows all possible execution paths in the Execution Model for *SEMS*. Note that this figure is also included in Appendix A for clearer visualization. The execution model consists of a total of 16 distinct paths for the execution, generated by the SAVE tool. This means that all possible probabilistic execution paths are generated with probabilities determined by choice operations. The criteria in Table 8 show that a patient with higher priority must be transported first to the hospital. Figure 12 shows the composition of probabilistic choice operations in terms of these criteria.

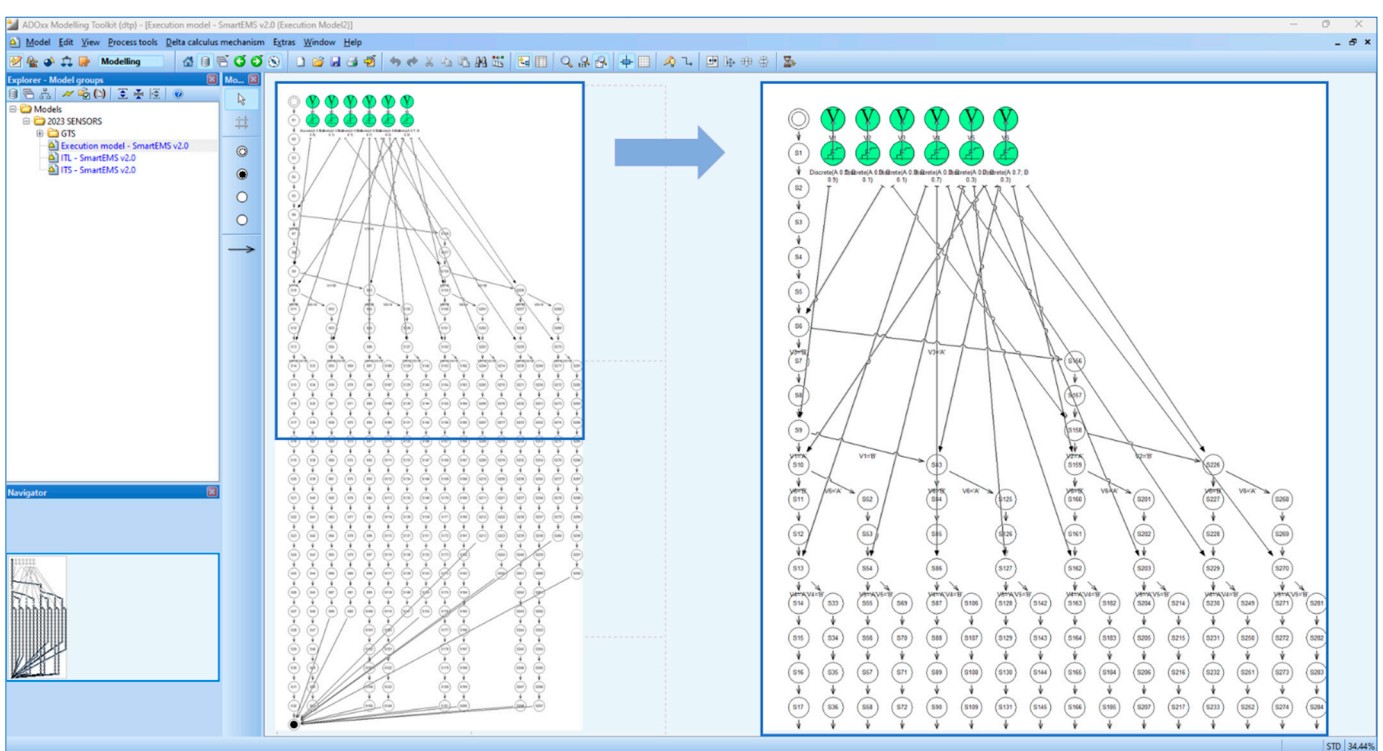

**Figure 11.** Execution model for SEMS.

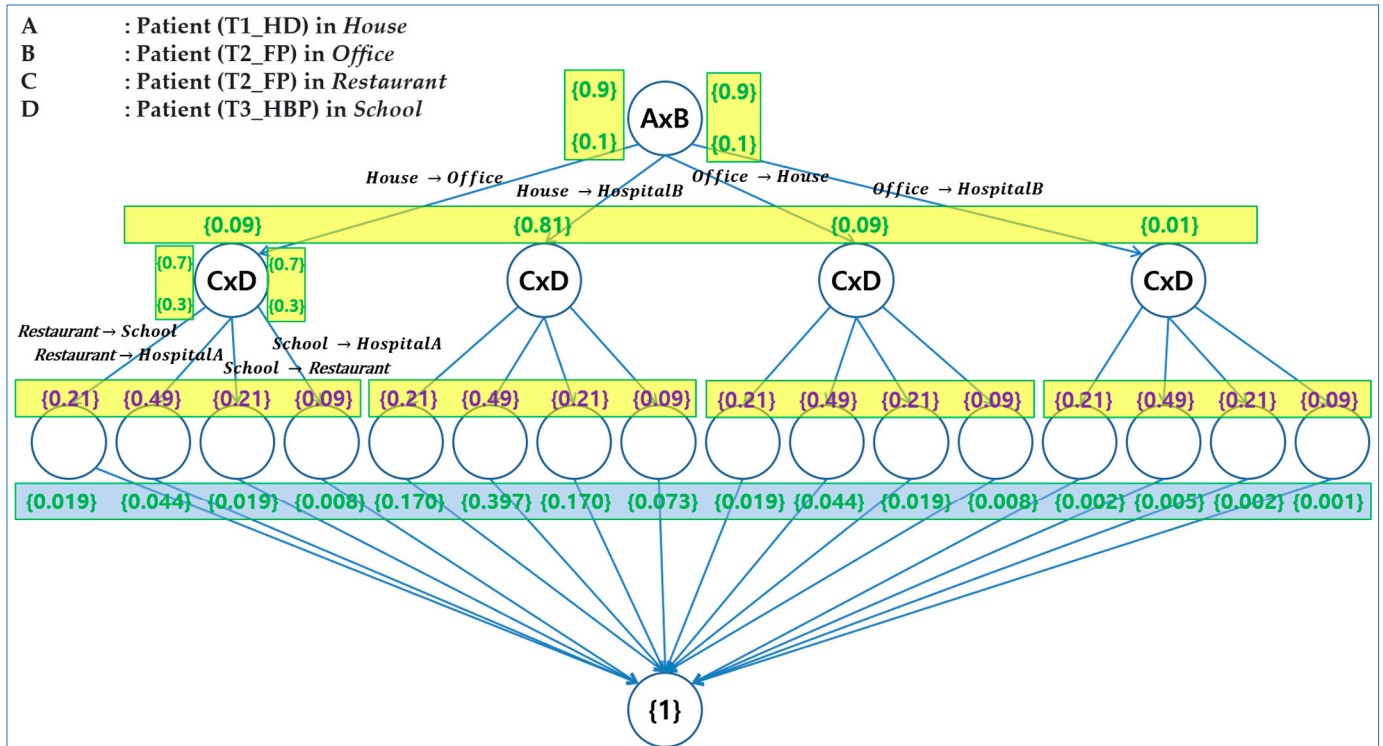

**Figure 12.** Safety and security requirements.

**Table 8.** Prioritized choice probabilities between *Patients*.

| Types of Illness | | Branch *A* | Branch *B* |
|---|---|---|---|
| *T*1 | *T*2 | 90% | 10% |
| *T*2 | *T*3 | 70% | 30% |
| *T*1 | *T*3 | 90% | 10% |
| Etc. | | 50% | 50% |

When *AmbulanceX* and *AmbulanceY* perform choice operations for the patients, the operations are determined by priority, with probabilities defined in Table 8. In *SEMS*, it is assumed that a higher priority patient has a higher probability than a lower priority patient. In addition, as *AmbulanceX* and *AmbulanceY* take patients, they make probabilistic choices, as shown in Table 9, generated by the criteria in Table 8.

**Table 9.** Probabilities of choice operations by *AmbulanceX* and *AmbulanceY*.

| Ambulance | Branch *A*, *B* | Former | Latter |
|---|---|---|---|
| *AmbulanceX* | *House* vs. *Office* | 90% | 10% |
| | *HosptialB* vs. *Office* | 90% | 10% |
| | *HosptialB* vs. *House* | 10% | 90% |
| *AmbulanceY* | *Restaurant* vs. *School* | 70% | 30% |
| | *HosptialA* vs. *School* | 70% | 30% |
| | *HosptialA* vs. *Restaurant* | 30% | 70% |

4.3.4. Safety and Security Requirements

In *SEMS*, all patients must be transported to designated hospitals within their deadlines. When multiple patients are transported, they have to follow the rules to satisfy the requirements of *SEMS*. The ambulance transports the patient with higher priority first. The definitions of the criteria for the safety requirements are as follows:

(1) *Req*1: The ambulance must satisfy the transport deadlines of the patients.
(2) *Req*2: The ambulance must transport the patient with higher priority first when there is competition among patients.
(3) *Req*3: The ambulance must transport the patients to their assigned hospitals.

Specifications derived from these safety requirements are shown in Table 10. Note that each requirement is specified with GTS Logic, visually represented in Figure 13. As stated, the logic is out of the scope of this paper and is briefly described in the next subsection.

**Table 10.** Safety requirements for SEMS.

| Req' # | Description and Predicate |
|---|---|
| *Req*1-1 | *AmbulanceX* must inform *HosptialB* through *TRANSFER_to_HospitalB* that *Patient*(*info_T1_HD*) will arrive |
| | $ACTION(AmbulanceX, Send, TRANSFER\_to\_HospitalB(info\_T1\_HD))$ |
| *Req*1-2 | *AmbulanceX* must inform *HosptialB* through *TRANSFER_to_HospitalB* that *Patient*(*info_T2_FP*) will arrive |
| | $ACTION(AmbulanceX, Send, TRANSFER\_to\_HospitalB(info\_T2\_FP))$ |
| *Req*1-3 | *AmbulanceY* must inform *HosptialA* through *TRANSFER_to_HospitalA* that *Patient*(*info_T2_FP*) will arrive |
| | $ACTION(AmbulanceY, Send, TRANSFER\_to\_HospitalA(info\_T2\_FP))$ |
| *Req*1-4 | *AmbulanceY* must inform *HosptialA* through *TRANSFER_to_HospitalA* that *Patient*(*info_T3_HBP*) will arrive |
| | $ACTION(AmbulanceY, Send, TRANSFER\_to\_HospitalA(info\_T3\_HBP))$ |

| Req' # | Description and Predicate |
|---|---|
| *Req*2-1 | In 10 time units, *AmbulanceX* must inform *HosptialB* through *TRANSFER_to_HospitalB* that *Patient*(*info_T1_HD*) will arrive |
| | $BEFORE((AmbulanceX, Send, TRANSFER\_to\_HospitalB(info\_T1\_HD)), 10)$ |
| *Req*2-2 | In 20 time units, *AmbulanceX* must inform *HosptialB* through *TRANSFER_to_HospitalB* that *Patient*(*info_T2_FP*) will arrive |
| | $BEFORE((AmbulanceX, Send, TRANSFER\_to\_HospitalB(info\_T2\_FP)), 20)$ |
| *Req*2-3 | In 20 time units, *AmbulanceY* must inform *HosptialA* through *TRANSFER_to_HospitalA* that *Patient*(*info_T2_FP*) will arrive |
| | $BEFORE((AmbulanceY, Send, TRANSFER\_to\_HospitalA(info\_T2\_FP)), 20)$ |
| *Req*2-4 | In 30 time units, *AmbulanceY* must inform *HosptialA* through *TRANSFER_to_HospitalA* that *Patient*(*info_T3_HBP*) will arrive |
| | $BEFORE((AmbulanceY, Send, TRANSFER\_to\_HospitalA(info\_T3\_HBP)), 30)$ |
| *Req*3-1 | The *AmbulanceX*'s informing to *HosptialB* through *TRANSFER_to_HospitalB* that *Patient*(*info_T1_HD*) will arrive must occur earlier than the *AmbulanceX*'s informing to *HosptialB* through *TRANSFER_to_HospitalB* that *Patient*(*info_T2_FP*) will arrive |
| | $TIME\begin{pmatrix} (AmbulanceX, Send, TRANSFER\_to\_HospitalB(info\_T1\_HD)), \\ (AmbulanceX, Send, TRANSFER\_to\_HospitalB(info\_T2\_HBP)), 'before' \end{pmatrix}$ |
| *Req*3-2 | The *AmbulanceY*'s informing to *HosptialA* through *TRANSFER_to_HospitalA* that *Patient*(*info_T2_FP*) will arrive must occur earlier than the *AmbulanceX*'s informing to *HosptialA* through *TRANSFER_to_HospitalA* that *Patient*(*info_T3_HBP*) will arrive |
| | $TIME\begin{pmatrix} (AmbulanceY, Send, TRANSFER\_to\_HospitalA(info\_T2\_FP)), \\ (AmbulanceY, Send, TRANSFER\_to\_HospitalA(info\_T3\_HBP)), 'before' \end{pmatrix}$ |

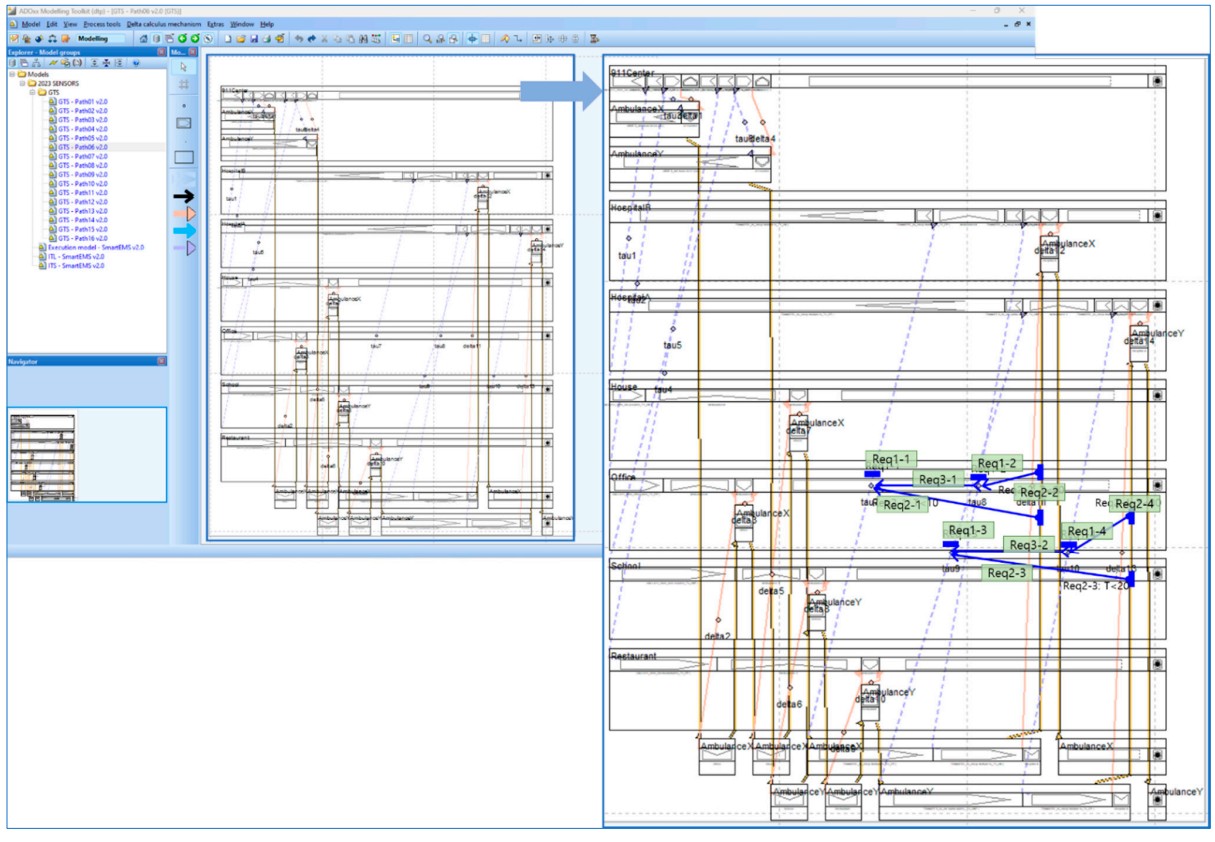

**Figure 13.** Meta snap of verification results in Execution Path 6.

4.3.5. Analysis of Probabilistic Verification for Requirements

In order to analyze the verification results for the requirements with probability, the following conditions are assumed:

(1) The total probability for paths that satisfy the safety requirements must be greater than 65%.

(2) The total probability for paths that satisfy only the deadline requirements must be greater than 95%.

Table 11 shows whether all the safety requirements are satisfied or not for all possible execution paths from Figures 11 and 12. The symbol 'O' signifies satisfaction, while 'X' denotes dissatisfaction.

**Table 11.** Analysis of probabilistic verification for SEMS.

| Path Req' # | 1 | 2 | 3 | 4 | 5 | 6 | 7 | 8 | 9 | 10 | 11 | 12 | 13 | 14 | 15 | 16 | Total |
|---|---|---|---|---|---|---|---|---|---|---|---|---|---|---|---|---|---|
| Prob. | 1.7 | 3.7 | 2.1 | 0.9 | 17.2 | 39.2 | 17.1 | 8.0 | 1.1 | 5.7 | 2.1 | 0.4 | 0.2 | 0.4 | 0.1 | 0.1 | 100 |
| *Req*1-1 | ○ | ○ | ○ | ○ | ○ | ○ | ○ | ○ | ○ | ○ | ○ | ○ | ○ | ○ | ○ | ○ | 100 |
| *Req*1-2 | ○ | ○ | ○ | ○ | ○ | ○ | ○ | ○ | ○ | ○ | ○ | ○ | ○ | ○ | ○ | ○ | 100 |
| *Req*1-3 | ○ | ○ | ○ | ○ | ○ | ○ | ○ | ○ | ○ | ○ | ○ | ○ | ○ | ○ | ○ | ○ | 100 |
| *Req*1-4 | ○ | ○ | ○ | ○ | ○ | ○ | ○ | ○ | ○ | ○ | ○ | ○ | ○ | ○ | ○ | ○ | 100 |
| *Req*2-1 | ○ | ○ | ○ | ○ | ○ | ○ | ○ | ○ | ✕ | ✕ | ✕ | ✕ | ✕ | ✕ | ✕ | ✕ | 64.8 |
| *Req*2-2 | ○ | ○ | ○ | ○ | ○ | ○ | ○ | ○ | ○ | ○ | ○ | ○ | ○ | ○ | ○ | ○ | 100 |
| *Req*2-3 | ○ | ○ | ○ | ○ | ○ | ○ | ○ | ○ | ○ | ○ | ○ | ○ | ○ | ○ | ○ | ○ | 100 |
| *Req*2-4 | ○ | ○ | ○ | ○ | ○ | ○ | ○ | ○ | ○ | ○ | ○ | ○ | ○ | ○ | ○ | ○ | 100 |
| *Req*3-1 | ○ | ○ | ○ | ○ | ○ | ○ | ○ | ○ | ✕ | ✕ | ✕ | ✕ | ✕ | ✕ | ✕ | ✕ | 64.8 |
| *Req*3-2 | ○ | ○ | ✕ | ✕ | ○ | ○ | ✕ | ✕ | ○ | ○ | ✕ | ✕ | ○ | ○ | ✕ | ✕ | 69.2 |

The execution paths with dissatisfaction are described in detail as follows:

(1) *Req*2-1: Paths with dissatisfaction for *Req*2-1 are Paths 9~16. As stated, *Req*2-1 is the requirement that *AmbulanceX* should transport patients within 10 time units. This dictates that the requirement for patients with *T*1 priority should be satisfied first.

(2) *Req*3-1: *Req*3-1 is the requirement that patients with *T*1 priority should be transported earlier than patients with *T*2 priority. Paths with dissatisfaction for *Req*3-1 are Paths 9~16. This shows that the requirement for patients with the priority *T*1 was not satisfied due to the fact that the patient with *T*2 priority was transported before those with *T*2 priority.

(3) *Req*3-2: *Req*3-2 is the requirement that patients with *T*2 should be transported earlier than patients with *T*3. Paths with dissatisfaction for *Req*3-2 are Paths 3, 4, 6, 7, 11, 12, 15, and 16. Among these, Paths 11, 12, 15, and 16 are the worst cases that do not satisfy requirements *Req*2-1, *Req*3-1, and *Req*3-2, and the sum of their probabilities is about 2.7%.

The results for the probabilities are generated by the Simulator of the SAVE tool, and the verification of requirements is also generated by the Analyzer and Verifier of the tool. Figure 13 shows the verification results as a GTS output with blue-colored arrows, generated for Path 6 of the Execution Model. Note that the figure is also included in Appendix A for clearer visualization.

As stated, GTS (Geo-Temporal Space) is a 2-dimensional graph used to formalize the visual representation of the system behavior of Smart IoT Systems in dTP-Calculus [18,19]. It consists of *System*, *Process*, and *Action* Blocks, and *Interaction* Edges of *Communication* and *Movement* among synchronous actions, as defined in dTP-Calculus. It shows the geographical relations among blocks in one dimension and the time-dependent synchronous relations for interactions of communication and movements among synchronous actions in another dimension. It is used to visually represent how a system executes itself in a specific geographical space at a specific time frame.

GTS-VL (Geo-Temporal Space-Visual Logic) is a first-order logic designed to formalize the visual representation of the requirements on GTS for Smart IoT Systems [18,19,21]. It follows the basic definition of first-order logic: terms and formulas. The main difference is the way of representing terms and formulas visually on GTS. Predicates are classified into the following categories:

(1) *Block* predicate: Represents the relations between System, Process, Action, and Interaction Blocks.
(2) *Temporal* predicate: Represents the temporal properties or relations for a block or among blocks.
(3) *Geo* predicate: Represents the geographical relations among blocks.
(4) *Interaction* predicate: Represents the occurrences of interactions.

Once the requirements are specified, the Verifier is responsible for checking whether each predicate is valid in the given GTS for the selected path of the EM of the systems or not. If a requirement is satisfied, the color of the graphical predicate is represented in blue; if not, in red. In Figure 13, all predicates are colored blue, implying visually that all the requirements in Table 10 are satisfied.

As a result of the verification for all the requirements of each path in Figure 12, Table 11 shows the summary of the verification. For example, safety requirements are satisfied in paths 1, 2, 5, and 6. Note that the ambulance transports T1-type patients to assigned hospitals within deadlines in paths 1, 2, 5, and 6.

In addition, paths 1 to 8 satisfy the deadlines for all patients (A, B, C, and D). The total probability of paths 1 to 8 is 89.9%. It is not appropriate for probabilistic requirements to have a total probability greater than 90%. Paths 1, 2, 5, and 6 satisfy all safety requirements. However, the total probability for Paths 1, 2, 5, and 6 is 61.8%, which does not satisfy the probabilistic requirement of the total probability needing to be greater than 65%. This means that the probabilities in Tables 8 and 9 have to be adjusted, as shown in Tables 12 and 13.

**Table 12.** Adjusted probability for Table 8.

| Types of Illness | | Branch *A* | Branch *B* |
|---|---|---|---|
| T1 | T2 | 98% | 2% |
| T2 | T3 | 70% | 30% |
| T1 | T3 | 98% | 2% |
| Etc. | | 50% | 50% |

**Table 13.** Adjusted probability for Table 9.

| Ambulance | Branch *A*, *B* | Former | Latter |
|---|---|---|---|
| *AmbulanceX* | *House* vs. *Office* | 98% | 2% |
| | *HosptialB* vs. *Office* | 98% | 2% |
| | *HosptialB* vs. *House* | 2% | 98% |
| *AmbulanceY* | *Restaurant* vs. *School* | 70% | 30% |
| | *HosptialA* vs. *School* | 70% | 30% |
| | *HosptialA* vs. *Restaurant* | 30% | 70% |

In Tables 12 and 13, the probability of Branch A for T1-type patients is increased from 90% to 98%, which means that prioritizing life-threatening patients has the most critical value for probability. As shown in the adjusted Table 14, the total probability from Path 1 to 8 is 98%, aligning appropriately with the probabilistic requirements for a total probability greater than 90%. In addition, the total probability for Paths 1, 2, 5, and 6 is 69.5%, meeting the probabilistic requirements for a total probability greater than 65%.

**Table 14.** Analysis of adjusted probabilistic verification for SEMS.

| Path | 1 | 2 | 3 | 4 | 5 | 6 | 7 | 8 | 9 | 10 | 11 | 12 | 13 | 14 | 15 | 16 | Total |
|------|-----|-----|-----|-----|------|------|------|-----|-----|-----|-----|-----|-----|-----|-----|-----|-------|
| Prob. | 0.5 | 0.9 | 0.5 | 0.2 | 19.7 | 48.4 | 19.9 | 7.9 | 0.3 | 1.0 | 0.5 | 0.2 | 0.0 | 0.0 | 0.0 | 0.0 | 100 |

The worst-case scenarios where all requirements, that is, *Req*2-1, *Req*3-1, and *Req*3-2, are not satisfied are Paths 11, 12, 15, and 16, and the summed probability of the cases is about 0.7% in total. Since the total probability of the worst cases was 2.7% before adjustment, it can be shown that the probability after adjustment was reduced by 2%.

In summary, this demonstrates how the analysis can influence results in controlling probability to meet the safety requirements of *SEMS*. *SEMS* is a representative example of Smart IoT Systems in Digital Twin.

It should be noted that the SAVE tool suite is freely accessible to the public and can be downloaded from the OMiLAB [27].

## 5. Proof of Concept

This section describes the proof of concept for the approach in this paper: SMES in Digital Twin.

The basic architecture for the proof of concept is shown in Figure 14. It demonstrates how the *SEMS* example in Digital Twin can be implemented in the real world with the SAVE tool on ADOxx for dTP-Calculus. Basically, the virtual part is implemented mainly with SAVE on ADOxx. However, there is additional information about the physical world, as well as the interface between the virtual and physical world. For the physical world, an experimental smart city platform was constructed to show the feasibility of how the smart city can be built and how the *SEMS* can be activated in the city for the application. Finally, the simulation of the *SEMS* by SAVE in the virtual world is triggered by the Digital Twin Scheduler, and the Smart IoT System for the *SEMS* in the physical world is activated accordingly, as commanded by the scheduler.

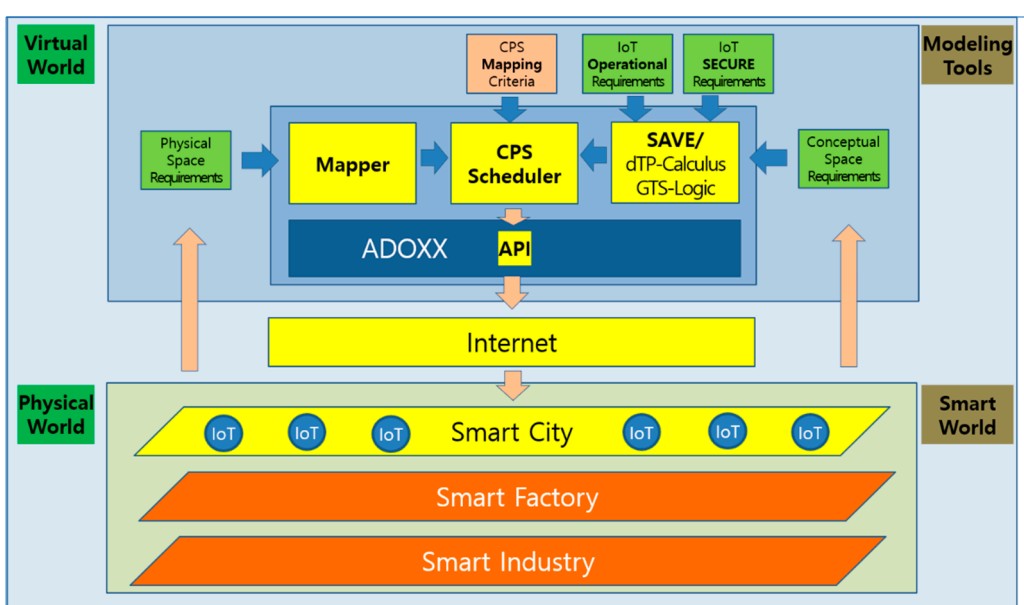

**Figure 14.** Architecture for the SEMS Example in Digital Twin.

### 5.1. Architecture

The architecture in Figure 14 consists of the following components:

(1)  Virtual World:

①  *SEMS* Virtual Requirements: SAVE

      (i)     *SEMS* Operational Requirements: Section 4.3.1

      (ii)    *SEMS* Safety/Security Requirements: Section 4.3.4

    ②    *SEMS* Physical Requirements: Mapper

(2)    Physical World:

    ①    City Map: A miniature of the target smart city consisting of:

      (i)     911 Center

      (ii)    Places

      (iii)   Hospitals

    ②    Smart IoT for Ambulance: Consists of the following devices and modules for the *SEMS*:

      (i)     Arduino: Plays a central role in controlling the IoT devices.

      (ii)    DC Motor: Controls the wheels of the IoT devices.

      (iii)   Line Tracer: Controls the direction of the IoT devices.

      (iv)   Wifi Module: Performs communication with the SAVE Scheduler on ADOxx in the virtual world.

    ③    Patients

(3)    SAVE Scheduler: An engine designed to schedule the processes of the *SEMS* using SAVE for smart IoT devices in the physical world. It generates commands for communication and movement actions of processes through the NodeJS server for interactions between the virtual world and the physical world.

(4)    API: All interactions to/from the smart IoT devices in the physical world are implemented in the JSON data format for communication from the virtual world.

### 5.2. Mapper

As stated, Mapper is a tool to model physical objects, that is, the smart IoT devices distributed across a smart city in the physical world within Digital Twin. This tool is developed on ADOxx to demonstrate the feasibility of the approach for the *SEMS* example in this paper, as part of the proof of concept. For the *SEMS* example, Figure 15 shows a city map where 911 Center, Places, Hospitals, and Patients and Doctors are distributed.

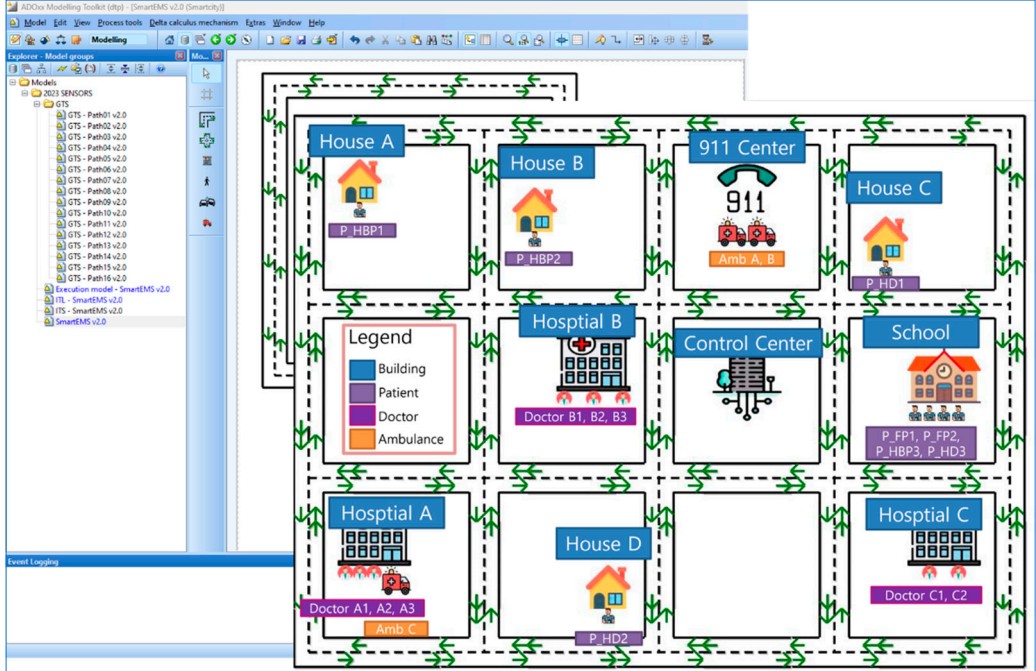

**Figure 15.** The SEMS map example.

### 5.3. SAVE Scheduler

As stated, the Scheduler is an engine to activate smart IoT devices in the physical world from the Simulator of the SAVE tool in the virtual world. Each process in SAVE is assigned to an IoT device in the *SEMS* System. Their communications and movements are triggered by the scheduler in real-time during simulation by the SAVE tool. An example of the API from the schedule to the IoT device is shown in Figure 16.

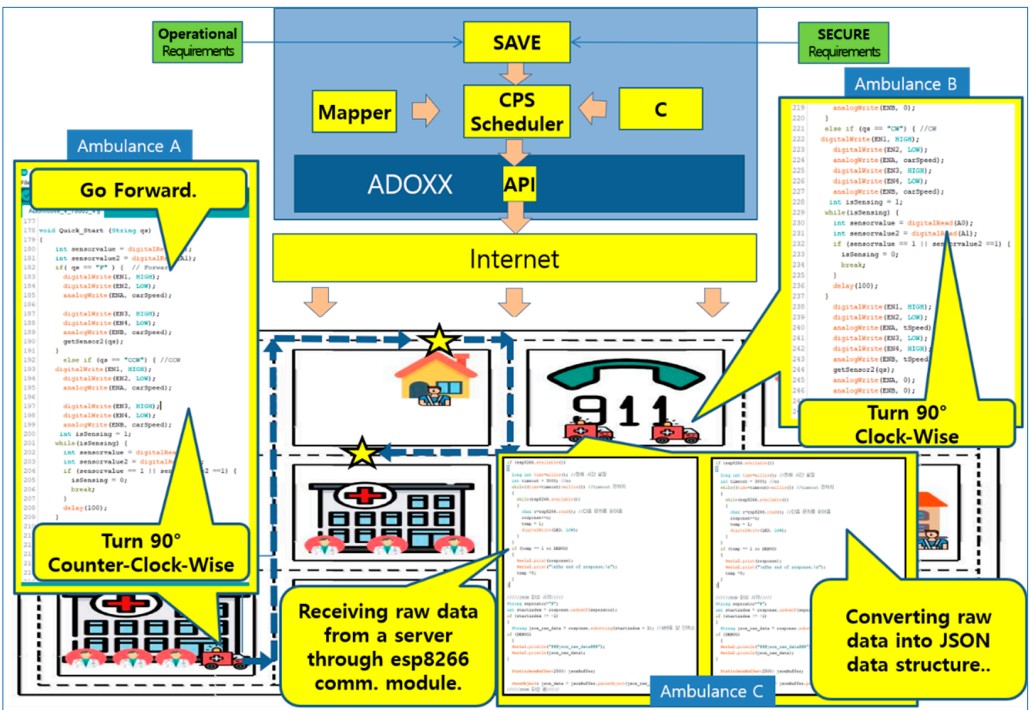

**Figure 16.** API Example from the Scheduler to the IoT Device in the SEMS example.

### 5.4. API

5.4.1. ADOxx Building Block

ADOxx Building Blocks are function blocks in ADOxx, whose functionalities are extended by importing external modules into the ADOxx platform. The building blocks supported by the current ADOxx platform from the ADOxx site are as follows [29]:

(1) Remote model documentation
(2) ADOxx web API
(3) ADOxx web dashboard
(4) LoLA Petri Net verification
(5) ADOxx web simulation
(6) Extended HTTP request

For the API between the SAVE scheduler and the smart IoT devices, the Extended HTTP Request Block has been utilized. The figure shows the data format for exchanging HTTP messages between a server and clients, where the request is the message from a client to a server to trigger an action from the server, and the response is the reply from the server to the client for the request. Here, the collection of the beginning line of the message and the HTTP header is called the request header, and, conversely, the payload of the message is called the request body.

The API from ADOxx utilizes the request message at the time of sending data to the server. The server receives the data stored in the HTTP request and sends the data to the target IoT device directly. For that, ADOxx provides the basic API, known as HTTP_REQUEST, for HTTP request communication.

### 5.4.2. NodeJS Communication Server

NodeJS is a platform used to develop network applications for extension. It guarantees high-processing performance for transactions through a single thread and an event loop by using JavaScript as an embedded language [30].

NodeJS works without other additional SW like Apache since it includes internal HTTP serve libraries. Consequently, NodeJS guarantees more control over web server operations.

More importantly, ADOxx provides developers for Digital Twin with such a service for communication between ADOxx and Arduino, as stated in the ADOxx Building Blocks.

### 5.4.3. Json Data Format

JSON (JavaScript Object Notation) is an open standard format for sending or receiving a data object, consisting of both key and value, in human-readable text format. It is known as a method to exchange data on the Internet. There is no limit to the types of data, and it is especially suitable for describing the values of parameters in computer programs [31].

Originally extended from JavaScript construct format, JSON has its own independent data format. In other words, since it is independent of programming languages and platforms, the codes for construct analysis and JSON data generation can be easily used in many programming languages.

In this paper, JSON is used as the basic data format for communication between ADOxx and NodeJS, as well as between NodeJS and Arduino.

### 5.4.4. Arduino

Arduino is an open-source electronic prototyping platform allowing users to develop interactive electronic devices. More specifically, it allows users to develop electronic objects that receive data from a number of switches and sensors and interact with an environment by controlling LED monitors and external electronic devices [32].

In this paper, a mobile IoT device is built on an Arduino board, installing a DC Motor, Line Tracer (TCRT500), Wifi module, etc., and is used as an Ambulance for the *SEMS* example in the physical world.

### 5.5. Activation of Smart IoT Devices in the Physical World

As stated, the real physical system in Digital Twin, based on Smart IoT Devices, is activated by the SAVE scheduler in the virtual world from SAVE and Mapper. A snapshot of the miniature of the smart city for the example is shown in Figure 17. It is the moment that two Ambulances are moving toward the places where their patients are located at House, Office, Restaurant, and School.

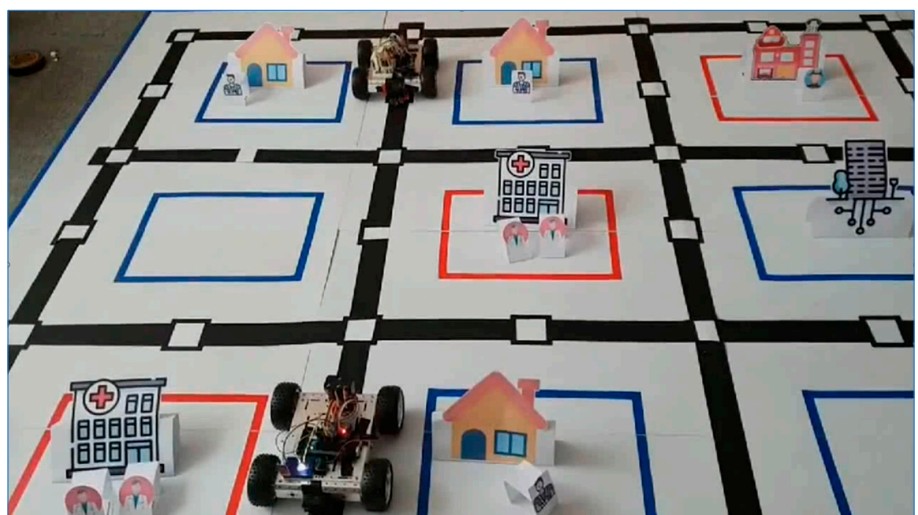

**Figure 17.** A snapshot of ambulances transporting patients in the SEMS system.

## 6. Comparative Study

This section presents a comparative study with related research, based on process algebra with probability, as follows:

(1)   PALOMA [33]: PALOMA allows the specification of geographical information for each agent in systems, using an exponential distribution probability model. It is useful for analyzing systems with distributed agents over geographical space by applying M2MAM (multi-class, multi-message Markovian Agent Models).

(2)   PACSR (Probabilistic ACSR) [34]: PACSR is a process algebra extended from ACSR (Algebra of Communicating Shared Resources) [35] with a probability property. It allows the specification of systems with probability for discrete distribution, particularly for the availability of the *resources* in the algebra. It defines transition rules for the following three actions only: timed actions, untimed events, and probabilistic actions. Based on the transition rule for the probabilistic action, it analyzes the possibility of using the resource needed for the action.

(3)   TPCCS (Timed Probabilistic CCS) [36]: TPCCS is a process algebra extended from CCS (Calculus of Communicating systems) [37] with a probability property. It provides transition rules for the composition of time and probability, and it allows simple probabilistic analysis through the definitions of probabilistic equivalences and bisimulations on processes.

The above process algebras allow only specific static probability models for specification of the systems. Further, there are limitations, such as PALOMA lacking the ability to specify operational requirements for systems, since it is not allowed to specify the mobility of the processes. In addition, there are no automatic functionalities for analyzing the probabilistic behaviors of systems with nondeterministic choice operations. Therefore, it is necessary to calculate them manually. Thus, the probability model can be utilized only in the specification step, but not in the analysis and verification steps.

Compared to the above process algebras, as shown in Table 15, dTP-Calculus allows the specification of systems with geographical inclusion relations, temporal restrictions, process mobilities, and nondeterministic choice operations with static and dynamic probability. With respect to the nondeterministic behavior of the systems, the calculus allows the specification of choice operations with a number of probability models, that is, Discrete, Normal, Exponential, and Uniform Models. It also supports the analysis of the nondeterministic probabilistic behavior of the systems based on the models, with the customized analysis service of the SAVE tool automatically performing the overload calculation of various nondeterministic choice operations with probabilities in the models. Most importantly, it supports the analysis and verification of probabilistic execution of the systems to determine the acceptance of the safety and security requirements of the systems based on dynamic probability.

**Table 15.** Comparison of process algebras.

| Process Algebra | Synchronization | Determinism | Probability Type | Probability Distribution | Time | Mobility | Visualization |
|---|---|---|---|---|---|---|---|
| dTP-Calculus | Synch | Deterministic and Nondeterministic | Static and Dynamic | Discrete Normal Exponential Uniform | ○ | ○ | ○ |
| PALOMA | Synch | Nondeterministic | Static | Exponential | ○ | × | × |
| TPCCS | Synch | Deterministic and Nondeterministic | Static | Discrete | ○ | × | × |
| PACSR | Synch | Deterministic and Nondeterministic | Static | Discrete | ○ | × | × |

In addition, SAVE allows the visual representation of Smart IoT Systems for Digital Twin in dTP-Calculus in different graphical views: System and Process views, as well as the execution model of the systems. Further, the simulated output of each possible execution path can be visualized on GTS, and all the safety and security requirements can be visually analyzed and verified on GTS. Consequently, all the specification, simulation, analysis, and verification processes can be consistently visualized in SAVE.

Most practically, the SAVE tool suite is available to the public as an open model in OMiLAB [27].

## 7. Conclusions and Future Research

This paper proposed an approach for analyzing the verification of safety and security requirements with static and dynamic probability using dTP-Calculus, and it also dealt with the unconditional nondeterministic behavior of Smart IoT Systems for Digital Twin. The approach was demonstrated with SEMS as a Digital Twin example in the SAVE tool. The SAVE tool demonstrated the applicability and feasibility of the approach. The SAVE tool can be very effective and efficient by visually generating all possible cases of the execution paths, namely the Execution Model. The paper also demonstrated how to control unsatisfied probabilistic requirements by adjusting or tuning the given dynamic probabilities. The utilization of dTP-Calculus makes the management of uncertainty possible through incorporating dynamic probability features within unconditional nondeterministic choice operations. In addition, a digital twin system for the SEMA example was constructed to demonstrate the feasibility of Digital Twin as a proof of concept. In this perspective, the SAVE tool can be used to apply the approach to real industrial applications of Smart IoT Systems in Digital Twin. The SAVE tool is an open model, which can be downloaded from OMiLAB [27].

Future theoretical research includes defining the basic notion of different probabilistic equivalences between groups of processes in dTP-Calculus, developing a method to tune the degree of dynamic probabilistic equivalences among the groups, and applying the method to control risk situations occurring due to the lack of probabilistic equivalences in terms of risk management in Smart IoT Systems for Digital Twin, among other aspects. Future research in practice includes demonstrating the efficiency and effectiveness of the approach by applying dTP-Calculus on SAVE to real Smart IoT industry cases in Digital Twin.

**Author Contributions:** Conceptualization, M.L. and J.S.; methodology, M.L.; software, J.S., S.L. and D.K.; validation, M.L., D.K. and J.S.; formal analysis, J.S. and S.L.; investigation, J.S. and S.L.; writing—original draft preparation, J.S. and S.L.; writing—review and editing, M.L. and D.K.; visualization, J.S. and S.L.; supervision, M.L.; project administration, M.L. All authors have read and agreed to the published version of the manuscript.

**Funding:** This work was supported by the National Research Foundation of Korea (NRF-2022K1A3A1A18079935).

**Institutional Review Board Statement:** Not applicable.

**Informed Consent Statement:** Not applicable.

**Data Availability Statement:** Data are contained within the article.

**Conflicts of Interest:** The authors declare that they have no competing interests.

## Appendix A

*Smart EMS* ::= *House* ‖ *Office* ‖ *Restaurant* ‖ *School* ‖ *911 Center*[*AmbulanceX* ‖ *AmbulanceY*] ‖ *HospitalA* ‖ *HospitalB*;

*House* ::= *CALL911_from_House*($\overline{info\_T1\_HD}$) · *AmbulanceX in* · *AmbulanceX out* · φ;

*Office* ::= *CALL911_from_Office*($\overline{info\_T2\_FP}$) · *AmbulanceX in* · *AmbulanceX out* · φ;

*Restaurant* ::= *CALL911_from_Restaurant*($\overline{info\_T2\_FP}$) · *AmbulanceY in* · *AmbulanceY out* · φ;

*School* ::= *CALL911_from_School*($\overline{info\_T3\_HBP}$) · *AmbulanceY in* · *AmbulanceY out* · φ;

*911 Center* ::= *CALL911_from_House*(*info_T1_HD*) · *CALL911_from_Office*(*info_T2_FP*) · *ORDER_AmbulanceX*($\overline{GO}$) · *put AmbulanceX* ·
　*CALL911_from_Restaurant*(*info_T2_FP*) · *CALL911_from_School*(*info_T3_HBP*) · *ORDER_AmbulanceY*($\overline{GO}$) · *put AmbulanceY* · φ;

*AmbulanceX* ::= *ORDER_AmbulanceX*(*GO*) · *911 Center put* ·
　(((*in House* · *out House* · (*TRANSFER_to_HospitalB*($\overline{info\_T1\_HD}$) · *in HospitalB* · *out HospitalB* · *in Office* · *out Office* · *TRANSFER_to_HospitalB*($\overline{info\_T2\_FP}$) · *in HospitalB* · *out HospitalB*)(0.1)(0.9) $+_D$
　(*in Office* · *out Office* · *TRANSFER_to_HospitalB*($\overline{info\_T2\_FP}$) · *in HospitalB* · *in House* · *out House* · *TRANSFER_to_HospitalB*($\overline{info\_T1\_HD}$) · *in HospitalB* · *out HospitalB*)(0.9))(0.1)) · φ;
　(*in House* · *out House* · *TRANSFER_to_HospitalB*($\overline{info\_T1\_HD}$) · *in HospitalB* · *out HospitalB*)(0.9) $+_D$
　(*in Office* · *out Office* · *TRANSFER_to_HospitalB*($\overline{info\_T2\_FP}$) · *in HospitalB* · *out HospitalB*)(0.1) $+_D$

*AmbulanceY* ::= *ORDER_AmbulanceY*(*GO*) · *911 Center put* ·
　(((*in Restaurant* · *out Restaurant* · (*TRANSFER_to_HospitalA*($\overline{info\_T2\_FP}$) · *in HospitalA* · *out HospitalA* · *in School* · *out School* · *TRANSFER_to_HospitalA*($\overline{info\_T3\_HBP}$) · *in HospitalA* · *out HospitalA*)(0.3))(0.7) $+_D$
　(*in School* · *out School* · *TRANSFER_to_HospitalA*($\overline{info\_T3\_HBP}$) · *in HospitalA* · *out HospitalA*)(0.3))(0.7)) $+_D$
　(*in School* · *out School* · (*TRANSFER_to_HospitalA*($\overline{info\_T3\_HBP}$) · *in HospitalA* · *out HospitalA* · *in Restaurant* · *out Restaurant* · *TRANSFER_to_HospitalA*($\overline{info\_T2\_FP}$) · *in HospitalA* · *out HospitalA*)(0.3)(0.7)) · φ;
　(*in Restaurant* · *out Restaurant* · *TRANSFER_to_HospitalA*($\overline{info\_T2\_FP}$) · *in HospitalA* · *out HospitalA*)(0.7) $+_D$

*HospitalA* ::= *TRANSFER_to_HospitalA*(*info_T2_FP*) \
　(*TRANSFER_to_HospitalA*(*info_T3_HBP*) · *AmbulanceY in* \ (*TRANSFER_to_HospitalA*(*info_T2_FP*) · *AmbulanceY in* · *AmbulanceY out* · φ;)
　· *AmbulanceY out* · *TRANSFER_to_HospitalA*(*info_T3_HBP*) · *AmbulanceY in* · *AmbulanceY out* · φ;)
　· *AmbulanceY in* \ (*TRANSFER_to_HospitalA*(*info_T3_HBP*) · *AmbulanceY in* · *AmbulanceY out* · φ;)
　· *AmbulanceY out* · *TRANSFER_to_HospitalA*(*info_T3_HBP*) · *AmbulanceY in* · *AmbulanceY out* · φ;

*HospitalB* ::= *TRANSFER_to_HospitalB*(*info_T1_HD*) \
　(*TRANSFER_to_HospitalB*(*info_T2_FP*) · *AmbulanceX in* \ (*TRANSFER_to_HospitalB*(*info_T1_HD*) · *AmbulanceX in* · *AmbulanceX out* · φ;)
　· *AmbulanceX out* · *TRANSFER_to_HospitalB*(*info_T1_HD*) · *AmbulanceX in* · *AmbulanceX out* · φ;)
　· *AmbulanceX in* \ (*TRANSFER_to_HospitalB*(*info_T2_FP*) · *AmbulanceX in* · *AmbulanceX out* · φ;)
　· *AmbulanceX out* · *TRANSFER_to_HospitalB*(*info_T2_FP*) · *AmbulanceX in* · *AmbulanceX out* · φ;

**Figure A1.** dTP-Calculus in Figure 9.

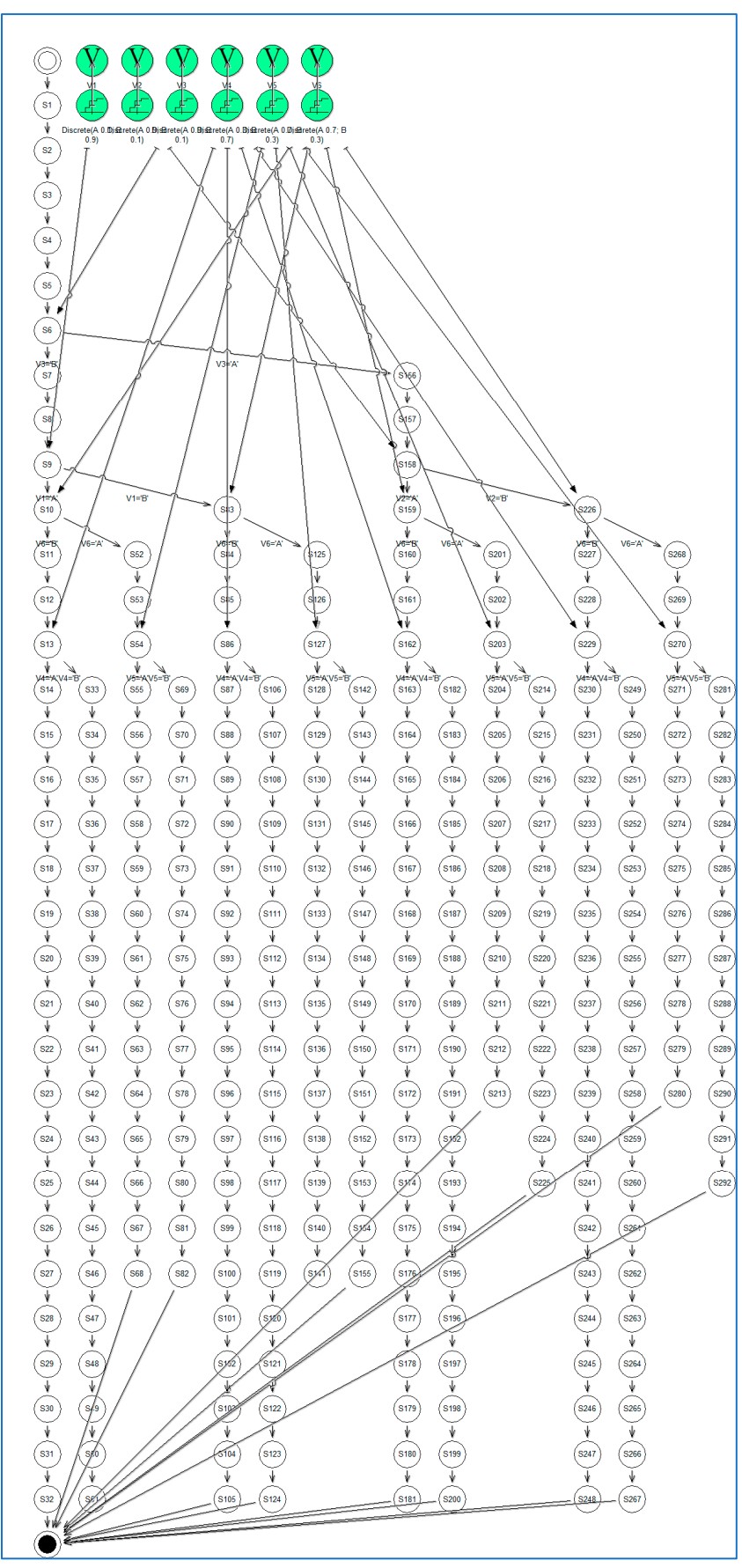

**Figure A2.** Execution Model in Figure 11.

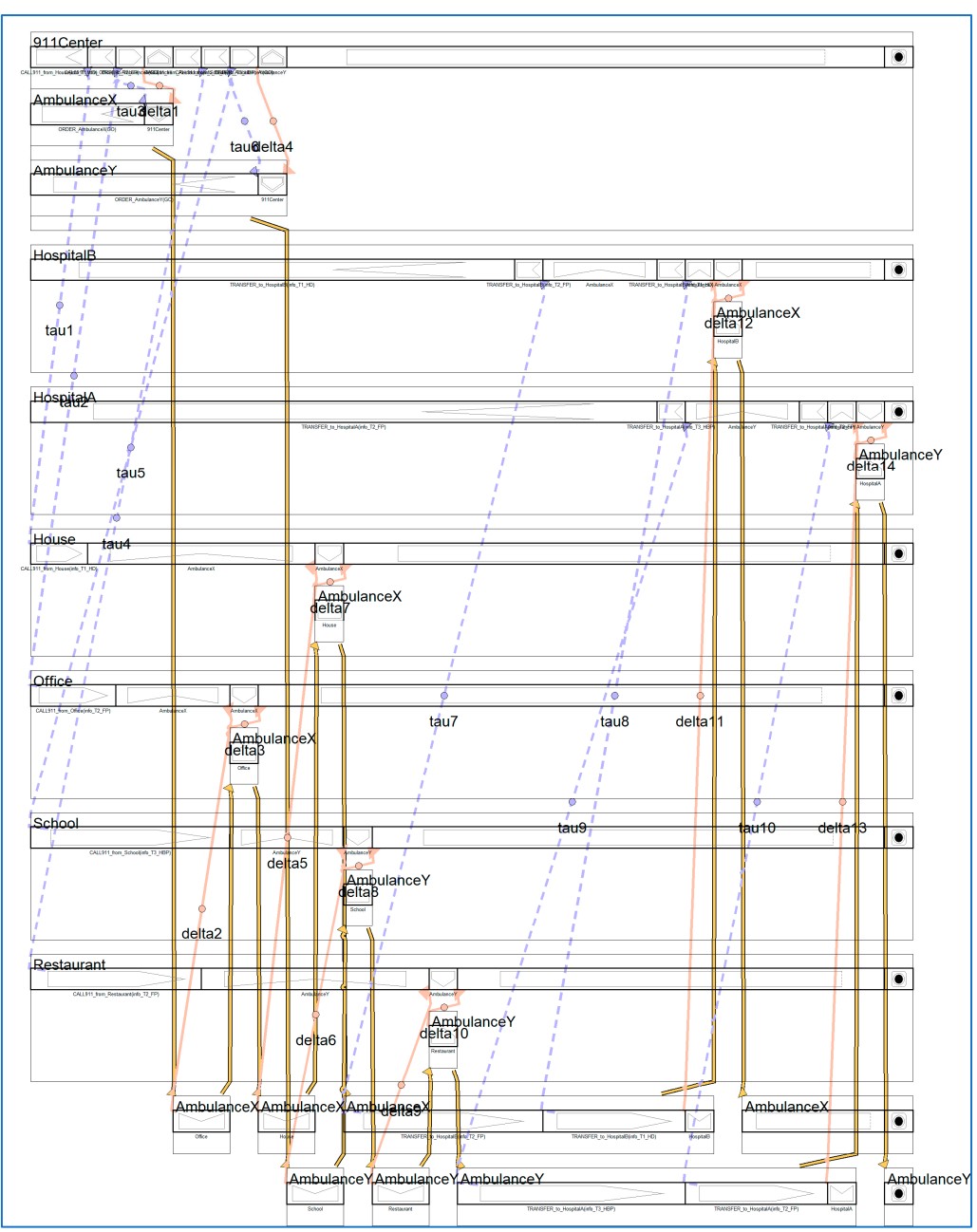

**Figure A3.** GTS Model in Figure 13.

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
