# Peer review of "Process Algebraic Approach for Probabilistic Verification of Safety and Security Requirements of Smart IoT (Internet of Things) Systems in Digital Twin"

_sensors, doi:10.3390/s24030767_

Round 1
Reviewer 1 Report
Comments and Suggestions for Authors
1. The introduction is overly lengthy and could benefit from being divided into subsections or shortened for brevity. It currently includes detailed discussions of various studies and findings; a concise summary highlighting the key points would enhance both readability and clarity.
2. There is no clear contribution in the paper.
3. The figures are not very clear.
4. It would be beneficial to clearly state the main objectives or hypotheses. This clarifies the rationale behind the chosen methods, setting a clear context for the study.
5. The presentation of the results, particularly in tables and figures, needs to be more apparent. Data must be presented in an easy way to interpret and understand.
6. A thorough discussion that interprets the findings in the context of the study's objectives and existing literature. More analysis of how the results align or differ from previous investigations and the implications of these findings is needed.
7. From section 2 onwards, the sections jump between tests and results without a straightforward narrative or logical flow. Establishing a cohesive storyline that connects all the tests and results would enhance the reader's comprehension.
8. If there are any novel or unexpected findings, these should be highlighted and discussed in depth. The significance of these findings in the field context should be clearly articulated.
9. a critical analysis of any limitations in the methodology that might affect the interpretation of the results.
10. References are inserted in the middle of the sentences and should be at the end or start.
11. A methodology section is required
12. Most sections are explained via bullet points; please use paragraphs.
13. The manuscript is quite lengthy.
14. Equations need to be numbered.
15. the description of the construct of the dTP-Calculus syntax is very confusing. A straightforward presentation method is required. (e.g. 9) Parallel: It defines that two processes in the parallel relationship are executed 252 at the same time, as shown in (9)). Is the (9) here a reference or what?? this is the case with all the points mentioned.
16. The conclusion section lists various findings, but it could benefit from a more structured summary that clearly articulates the main conclusions drawn from the research.The conclusions should be directly tied back to the initial objectives or hypotheses of the study. It's important to state how the findings address these objectives or what new understanding they bring to the field. The section should critically analyze the results, discussing their significance in existing literature and the study's contribution to the field. This includes discussing any novel findings or contradictions to previous investigations.
Comments on the Quality of English Language
A thorough English proofreading is required.
Reviewer 2 Report
Comments and Suggestions for Authors
Some figures and tables are unnecessary and hard to read. Please use the attached document to update them in the manuscript

Comments on the Quality of English LanguageExtensive editing to fix grammatical errors are required. Please follow the suggestions in the attached document.
Reviewer 3 Report
Comments and Suggestions for Authors
The paper concerns probabilistic verification of safety and security requirements of smart IoT systems in digital twins using process algebras, also called ‘process calculi’. The authors propose a new process algebra – dTP-Calculus – that can help to specify the nondeterministically predictable behaviour with probability and verify the safety and security requirements of Smart IoT Systems. The article is relevant and well written; the results are novel and make a favourable impression. Nevertheless, we should point out some shortcomings.
1. Some references are listed without DOI, so it is hard to find them. For example, in which journal paper [6] was published or maybe [6] is a book? Citations from Wikipedia also look somewhat strange in a scientific article. I recommend replacing them with more reliable sources.
2. The bibliography review poorly reflects classical publications on process calculi (or process algebra). For example, the works by such a classic as Robert Milner are not mentioned.
3. Some figures are difficult to read (see Fig. 5, 6, 10, 16). Moreover, it is almost impossible to read the text in Fig. 7, 11, 13, and 17, even at maximum resolution, which obstructs the perception of the material.
4. The conclusions are too brief and need expanding. It would be interesting to see the practical significance of the results obtained.
Nevertheless, the article is interesting and deserves publication, and the comments made can be corrected without re-reviewing.
Comments on the Quality of English Language
The quality of English language is acceptable. Nevertheless, I recomend the authors to check the paper carefully.
Round 2
Reviewer 1 Report
Comments and Suggestions for Authors
Accept in present form
Reviewer 2 Report
Comments and Suggestions for Authors
Authors addressed my comments.